# The Ets protein Pointed P1 represses Asense expression in type II neuroblasts by activating Tailless

**Rui Chen, Xiaobing Deng, Sijun Zhu** *

Department of Neuroscience and Physiology, State University of New York Upstate Medical University, Syracuse, New York, United States of America

* zhus@upstate.edu

## Abstract

Intermediate neural progenitors (INPs) boost the number and diversity of neurons generated from neural stem cells (NSCs) by undergoing transient proliferation. In the developing *Drosophila* brains, INPs are generated from type II neuroblasts (NBs). In order to maintain type II NB identity and their capability to produce INPs, the proneural protein Asense (Ase) needs to be silenced by the Ets transcription factor pointed P1 (PntP1), a master regulator of type II NB development. However, the molecular mechanisms underlying the PntP1-mediated suppression of Ase is still unclear. In this study, we utilized genetic and molecular approaches to determine the transcriptional property of PntP1 and identify the direct downstream effector of PntP1 and the *cis*-DNA elements that mediate the suppression of *ase*. Our results demonstrate that PntP1 directly activates the expression of the transcriptional repressor, Tailless (Tll), by binding to seven Ets-binding sites, and Tll in turn suppresses the expression of Ase in type II NBs by binding to two hexameric core half-site motifs. We further show that Tll provides positive feedback to maintain the expression of PntP1 and the identity of type II NBs. Thus, our study identifies a novel direct target of PntP1 and reveals mechanistic details of the specification and maintenance of the type II NB identity by PntP1.

## Author summary

Type II neuroblasts (NBs) are the neural stem cells (NSCs) in *Drosophila* central brains that produce neurons by generating intermediate neural progenitors (INPs) to boost brain complexity, as mammalian NSCs do during the development of neocortex. The key to the generation of INPs from type II NBs is the suppression of proneural protein Asense (Ase) in type II NBs by the Ets family transcription factor Pointed P1 (PntP1), but how PntP1 suppresses Ase expression remains unclear. In this study, we provided evidence to demonstrate that PntP1 directly activates the orphan nuclear receptor Tailless (Tll), which in turn suppresses Ase expression to maintain the capability of type II NBs to produce INPs. Meanwhile, Tll provides positive feedback to maintain the expression of PntP1 and type II NB identity. We further identified seven PntP1 binding sites in the *tll* enhancer regions and two Tll binding sites in the *ase* regulatory regions that mediate the activation

**Data Availability Statement:** All relevant data are within the manuscript and its Supporting Information files.

**Funding:** This work was supported by the National Institute of Neurological Disorders and Stroke of

the National Institutes of Health under Award
Numbers R01NS085232 (S.Z.) and R21NS109748
(S.Z.). The funders did not play any roles in the
design, data collection and analysis, decision to
publish, or preparation of the manuscript of this
study.

**Competing interests:** The authors have declared
that no competing interests exist.

of *tll* and the suppression of *ase*, respectively. Our work reveals detailed mechanisms of
the specification and maintenance of the type II NB identity by PntP1.

## Introduction

Constructing a functional nervous system requires generation of a large number of diverse
types of neurons and glia during development. Transient amplifying neural progenitors play
pivot roles in boosting the number and diversity of neurons or glia generated from neural
stem cells (NSCs). In *Drosophila* larval central nervous system (CNS), one type of neural pro-
genitors, called intermediate neural progenitors (INPs) is exclusively produced by a specific
type of NSCs, called type II neuroblasts (NBs), which were discovered over a decade ago [1–3].
Compared to other types of NBs in *Drosophila* CNS, including type 0 NBs that directly pro-
duce neurons at embryonic stages [4–6] and type I NBs that produce neurons by generating
terminally dividing ganglion mother cells (GMCs) [7,8], type II NBs produce at least 4 times
more neurons with more diverse subtypes by generating INPs, which undergo several rounds
of self-renewing divisions to produce GMCs [1,9–13].

Type II NBs differ from type I and type 0 NBs by the lack of the expression of the proneural
protein Asense (Ase). Although the exact role of Ase in type 0 and type I NBs remains unclear,
the absence of Ase in type II NBs is a prerequisite for the generation of INPs [3,10]. Ectopic
Ase expression in type II NBs leads to transformation of type II NBs to type I NBs, depletion of
INPs, and direct production of GMCs from the NBs [3,10]. Our previous studies have shown
that Pointed P1 (PntP1), which is a member of the E26 transformation-specific (Ets) family, is
a master regulator of type II NB specification and is responsible for the suppression of Ase
[9,10]. Misexpression of PntP1 is sufficient to transform type I NBs into type II NBs by sup-
pressing Ase expression and promote the production of INPs. The transformed type II NB lin-
eages are equally susceptible to tumorigenic overproliferation resulting from loss of tumor
suppressors such as Brain Tumor (Brat) and Earmuff (Erm), the latter of which is a direct
downstream target of PntP1 in immature INPs (imINPs) [10,14–16]. However, it is not clear if
PntP1 acts as a transcriptional repressor to directly suppress Ase expression or acts as a tran-
scriptional activator to suppress Ase indirectly by activating the expression of other transcrip-
tion factor(s).

In this study, we showed that PntP1 functions as a transcriptional activator to indirectly
suppress Ase expression in type II NBs by activating the expression of the transcriptional
repressor Tailless (Tll). We provide genetic and molecular evidence to demonstrate that Tll is
a direct target of PntP1 in type II NBs and identified seven PntP1 binding sites that mediate
the activation of *tll*, whereas Tll directly suppresses Ase expression by binding to two hexame-
ric core half-site motifs. We further show that Tll also provides positive feedback to maintain
the expression of PntP1 and the identity of type II NBs. Therefore, our work identifies a novel
direct target of PntP1 and elucidates mechanistic details of PntP1-mediated specification and
maintenance of the type II NB identity.

## Results

### Identification of *cis*-elements mediating the suppression of *ase* in type II NBs

Our previously studies showed that the misexpression of the Ets repressor Yan could antago-
nize PntP1's function and lead to ectopic Ase expression in type II NBs, suggesting that PntP1

unlikely functions as a transcriptional repressor to directly suppress Ase expression in type II NBs [10]. Therefore, we hypothesized that a transcriptional repressor acts downstream of PntP1 to suppress Ase through directly binding to *ase* regulatory regions in type II NBs. To identify such a repressor, we took a bottom-up approach by mapping *cis*-elements in the *ase* regulatory region that mediate *ase* suppression in type II NBs so that the *cis*-elements could be used to identify the transcriptional repressor using bioinformatical, biochemical, and/or genomic approaches. To identify the *cis*-elements, we adopted a well-established enhancer-activity-assay [17] by inserting the *ase* enhancer fragment upstream of the *Drosophila* synthetic core promoter (DSCP) to drive GAL4 transcription in the *pBPGUW* vector (Fig 1A). Then the expression pattern of the GAL4 reporter lines, which are named *pDes-(ase)enhancer-GAL4*, was examined by driving the expression of mCD8-GFP under the control of upstream activating sequence (UAS) (*UAS-mCD8-GFP*).

Previous studies have shown that two overlapping *ase* enhancer segments, the *F:2.0* fragment [18] and the *GMR20B05* fragment [17], activate reporter expression only in type I NBs but not type II NBs, suggesting that the overlapping region contains the *cis*-repressive element. The overlapping region is a 675-base pair (bp) fragment located at -1,734bp ~ -1,060bp from the transcription start site (TSS) of *ase* (Fig 1A). Indeed, this 675-bp enhancer fragment drove the GAL4 transgene expression in all type I NBs but not in any type II NBs when it was inserted into the GAL4 reporter construct (Fig 1B, 1H and 1W). To map the *cis*-repressive element, we then made a series of deletions of the 675-bp fragment and used these truncated fragments for constructing the *pDes-(ase)enhancer-GAL4* reporter lines. By examining the expression of these GAL4 reporter lines, we found that deleting the region from -1,295bp to -1,280bp led to activation of the GAL4 in all type I and type II NBs (Fig 1C–1H and 1W), indicating that this 16-bp region contains the *cis*-repressive element. Further, our GAL4 reporter lines also showed that deleting the region from -1,138bp to -1,086bp resulted in a complete loss of GAL4 expression in all NBs in the larval brains (Fig 1I–1W), indicating that this region contains a *cis*-active element that is essential for activating *ase* expression in all NBs.

## Tll suppresses the expression of *pDes-(ase)enhancer-GAL4* reporters containing the *cis*-repressive element

Based on the sequence of the mapped 16-bp region that contains the repressive element, then we predicted potential binding transcription factors using a combination of the programs MEME (Mutiple Em for Motif Elicitation, https://meme-suite.org/meme/tools/meme) [19] and Tomtom (https://meme-suite.org/meme/tools/tomtom) [20]. We identified a total of 83 candidates that could potentially bind to the 16-bp sequence, including the transcriptional repressor Tailless (Tll), which has been shown recently to repress Ase expression in type II NBs [21]. Therefore, we next asked whether Tll binds to the 16-bp repressive element to suppress *ase* expression. To this end, we first compared the suppression of Ase in type I NBs by the expression of *UAS-Tll* driven by *pDes-(ase)enhancer-GAL4* reporter lines with or without the 16-pb *cis*-repressive element. The rational is that if Tll suppresses Ase expression through interacting with the repressive element, the misexpressed Tll would exert negative feedback inhibition of the GAL4 transgenes by binding to the repressive element. Subsequently, the expression of *UAS-Tll* driven by *pDes-(ase)enhancer-GAL4* containing the repressive element would be relatively weak and the expression of Ase may not be suppressed (Fig 2A). In contrast, when *UAS-Tll* is driven by *pDes-(ase)enhancer-GAL4* lines that do not contain the repressive element, the expression of the GAL4 transgenes would be stronger due to the lack of the negative feedback inhibition. Consequently, the expression of *UAS-Tll* would also be strong, which would lead to suppression of Ase in the NBs (Fig 2B). Indeed, our results showed that

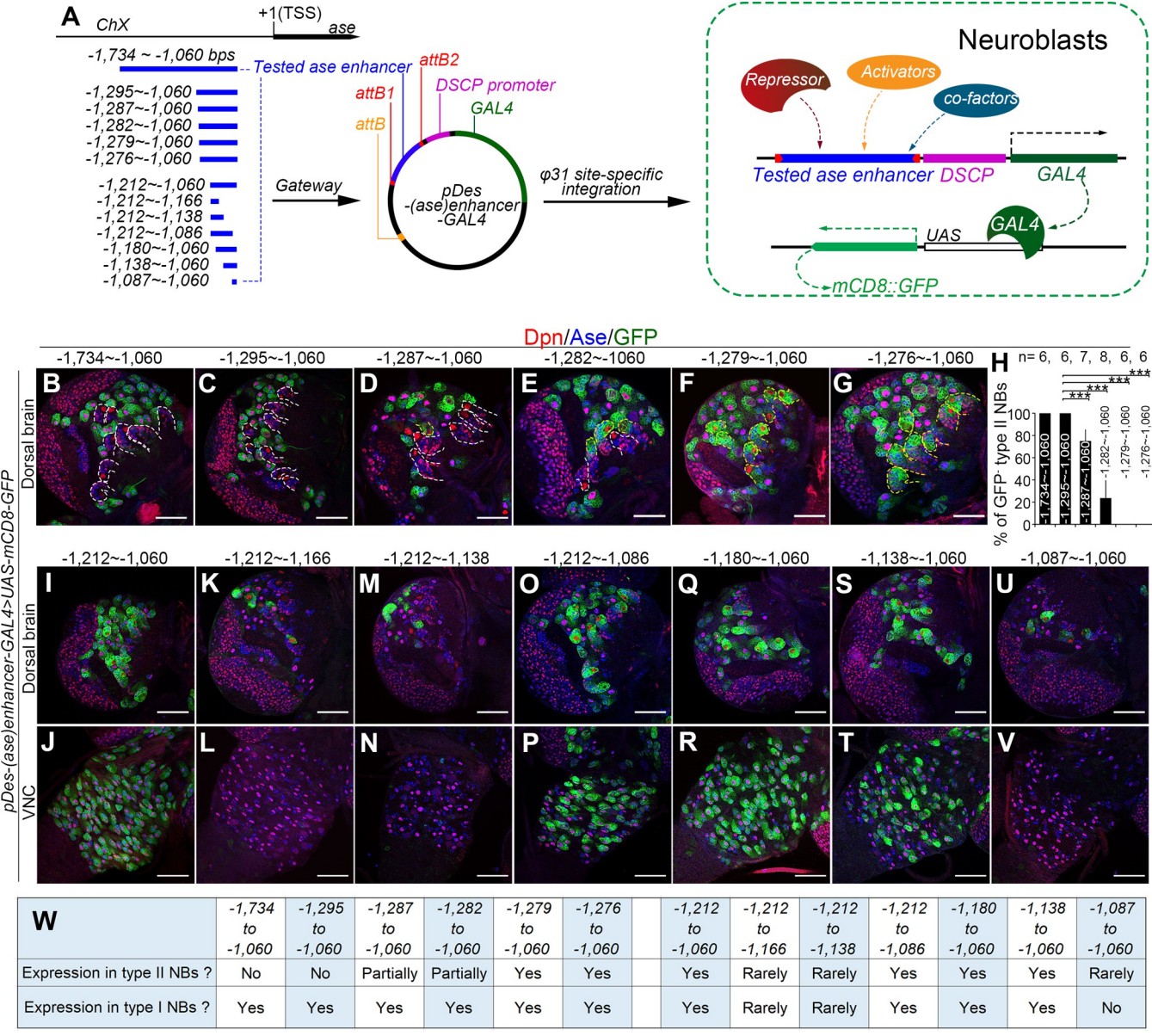

**Fig 1. Identification of *cis*-elements mediating the repression and activation of Ase expression.** (A) A diagram shows the tested *ase* enhancer fragments (in blue) and the strategy of the enhancer-activity-assay used to determine the activity of the fragments in type I NBs or in type II NBs. The *ase* enhancer fragments are amplified and cloned to upstream of *Drosophila* synthetic core promoter (DSCP)-GAL4 cassette in *pBPGUW* vector and the resulted *pDes-(ase)enhancer-GAL4* constructs are integrated into *Drosophila* genome for determining their expression in type I and type II NBs by driving the expression of the *UAS-mCD8-GFP* reporter. (B-G, I-V) Expression of mCD8-GFP driven by the *pDes-(ase)enhancer-GAL4* constructs with indicated *ase* enhancer fragments in type I NBs (Ase⁺ Dpn⁺, appearing magenta in the nucleus) or type II NBs (Ase⁻ Dpn⁺, appearing red in the nucleus) in the dorsal brains (B-G, I, K, M, O, Q, S, and U) or VNCs (J, L, N, P, R, T, and V). Images in (B-G) are for mapping the *cis*-elements mediating the suppression of *ase* in type II NBs and images in (I-V) are for mapping the *cis*-elements involved in activating *ase* in all NBs. Note that progressive deletion of the sequence from -1,295bp to 1,279bp leads to activation of mCD8-GFP in a subset of (D-E) or all type II NBs (F-G), whereas deletion of the sequence from -1,138bp to -1,086bp results in the loss of mCD8-GFP expression in all NBs in the VNC (L, N, and V) and in the majority of NBs in the brains (K, M, and U). Yellow dashed lines in (B-G) outline type II NB lineages with mCD8-GFP expression and white dashed lines in (B-G) outline the type II NB lineages without mCD8-GFP expression. In this and all the following figures, only images from one brain lobe or the thoracic segments of VNCs are shown. The brain lobes are oriented so that the midline is to the right and the anterior side of the brain up. Scale bars equal 50μm. (H) Quantifications of the percentage of type II NBs without mCD8-GFP expression in the brains with the GAL4 transgenes under the control of indicated *ase* enhancer fragments. Values of the bars are mean ± SD. ***, *P* < 0.001. (W) Summary of the expression of mCD8-GFP driven by the *pDes-(ase)enhancer-GAL4* constructs with indicated *ase* enhancer fragments in type I NBs or type II NBs.

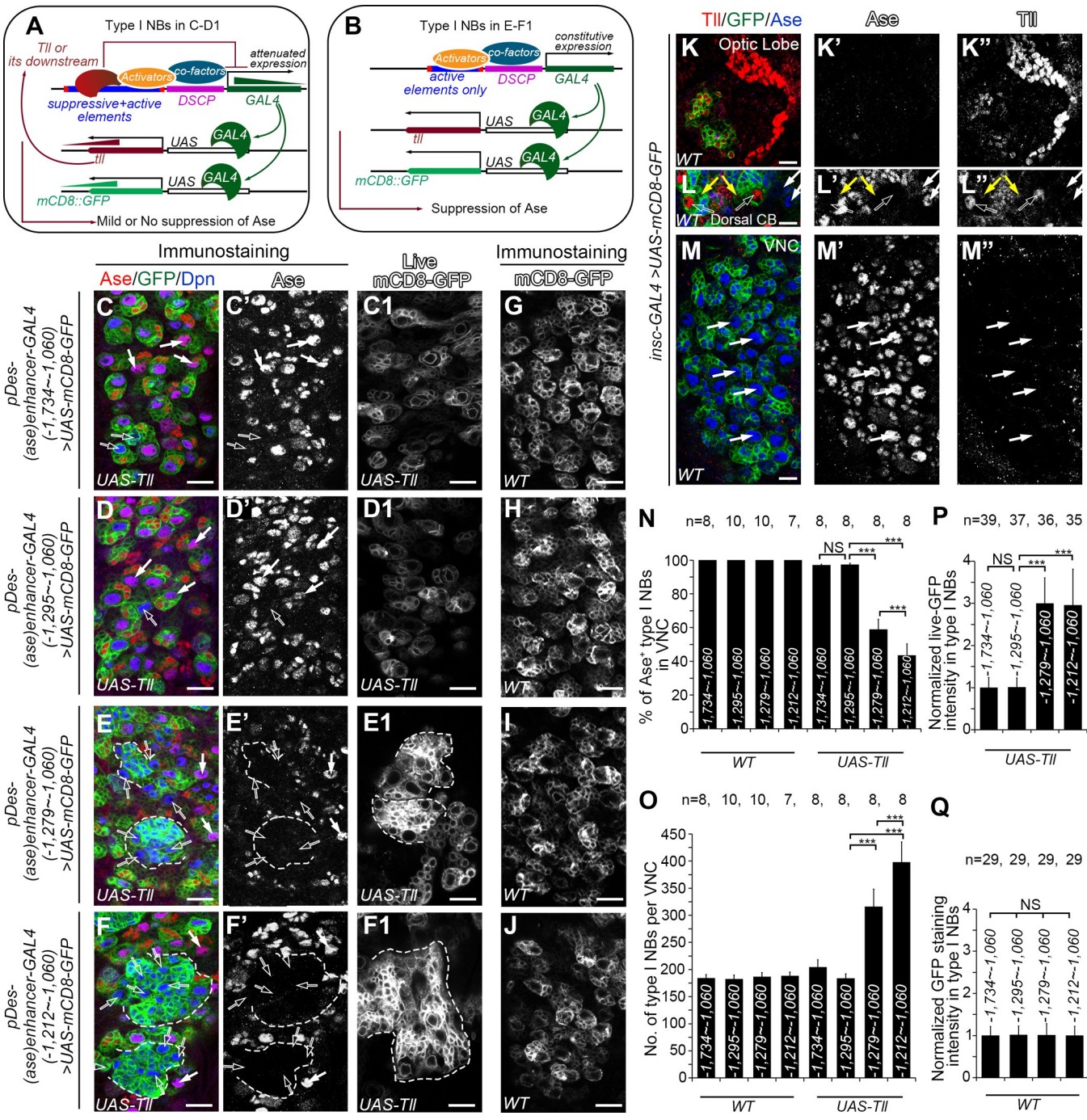

**Fig 2.** *UAS-Tll expression driven by pDes-(ase)enhancer-GAL4 is compromised when the repressive element is included in the GAL4 promoter.* In images (C-F) and (G-J) type I NBs are labeled with mCD8-GFP (in green) driven by *pDes-(ase)enhancer-GAL4* drivers containing indicated *ase* enhancer fragments, and counterstained with anti-Ase (in red) or anti-Dpn (in blue) antibodies. In images (C1-F1), type I NBs are imaged live with the same settings for comparing mCD8-GFP expression levels. In images (L-M) both type I and type II NBs are labeled with mCD8-GFP (in green) driven by *insc-GAL4*, and counterstained with anti-Tll (in red) or anti-Ase (in blue) antibodies under the same staining condition and imaging settings as those in the image (K). Scale bars equal 10μm. (A-B) Schematic diagrams show how the suppression of Ase by Tll is compromised [A, corresponding to (C-D1)] or maintained [B, corresponding to (E-F1)] in type I NBs. *pDes-(ase)enhancer-GAL4* first drives the expression of Tll. The resulted exogenous Tll proteins bind to the repressive element in the *pDes-(ase) enhancer-GAL4* driver and attenuates the subsequent GAL4 expression, leading to attenuated mCD8-GFP and Tll expression levels. The attenuated Tll expression is not adequate to suppress Ase expression (A). In contrast, *pDes-(ase)enhancer-GAL4* drivers containing only the active elements constitutively drive GAL4 expression, which results in higher expression levels of mCD8-GFP and Tll and subsequent suppression of Ase by Tll (B). (C-D') Expressing *UAS-Tll* driven by *pDes-(ase)enhancer-GAL4* with the *ase* enhancer fragment -1,734 ~ -1,060 bps (C-C') or -1,295 ~ -1,060 bps (D-D'), which contains both the repressive and active elements, leads to the suppression of Ase only in a few type I NBs (open arrows). The majority of type I NBs are still Ase+ (white arrows).

(E-F') Expressing *UAS-Tll* driven by *pDes-(ase)enhancer-GAL4* with the *ase* enhancer fragment -1,279 ~ -1,060 bps (E-E') or -1,212 ~ -1,060 bps (F-F'), which contains the active elements only, leads to suppression of Ase in an average of 42% or 57% of type I NBs (open arrows), respectively, and the formation of clusters of supernumerary type I NBs (dashed lines). (C1-F1) mCD8-GFP is expressed at relatively low levels in type I NBs when *UAS-Tll* is driven by *pDes-(ase)enhancer-GAL4* containing the *ase* enhancer fragment -1,734 ~ -1,060 bps (C1) or -1,295 ~ -1,060 bps (D1), but at much higher levels when *UAS-Tll* is driven by *pDes-(ase)enhancer-GAL4* containing the *ase* enhancer fragment -1,279 ~ -1,060 bps (E1) or -1,212 ~ -1,060 bps (F1). (G-J) mCD8-GFP is expressed at similar levels in type I NBs when *pDes-(ase)enhancer-GAL4* containing the *ase* enhancer fragment -1,734 ~ -1,060 bps (G), -1,295 ~ -1,060 bps (H), -1,279 ~ -1,060 bps (I) or -1,212 ~ -1,060 bps (J) is used as drivers in wild type VNCs. (K-K") The robust expression of Tll in optic lobe is detected by anti-Tll antibody (red). (L-L") Tll expression is detected robustly in type II NBs (open arrows) and weakly in newly born imINPs (yellow arrows), but not in type I NBs (white arrows) in the dorsal central brain (CB). (M-M") Tll expression is not detected in type I NBs (white arrows) in VNCs. (N-Q) Quantifications of the percentage of Ase+ type I NBs in VNCs (N), the number of total type I NBs per VNC (O), and normalized mCD8-GFP intensity in type I NBs (P, Q) in the VNCs with indicated genotypes. The average mCD8-GFP expression driven by the *pDes-(ase)enhancer-GAL4* with the fragment -1,734 ~ -1,060 bps in both the *WT* and *UAS-Tll* groups is normalized to 1. Values of the bars are mean ± SD.***, $P < 0.001$. NS, not significant.

expression of *UAS-Tll* driven by two *pDes-(ase)enhancer-GAL4* lines with the fragments -1,734 ~ -1,060 bps or -1,295 ~ -1,060 bps that contain the repressive element resulted in Ase suppression only in less than 3% of type I NBs (Fig 2C–2D' and 2N) and very subtle overproliferation of type I NBs (Fig 2C–2D' and 2O). However, misexpression of *UAS-Tll* using two *pDes-(ase) enhancer-GAL4* lines with the fragments -1,279 ~ -1,060 bps or -1,212 ~ -1,060 bps that do not contain the repressive element led to loss of Ase in more than 40% of type I NBs (Fig 2E–2F' and 2N) and a drastic increase in the number of NBs, which often clustered together, in the ventral nerve cord (VNC) (Fig 2E–2F' and 2O). Consistently, when *UAS-Tll* was co-expressed, the expression of *UAS-mCD8-GFP* driven by the *pDes-(ase)enhancer-GAL4* lines containing the repressive element was much weaker than that driven by *pDes-(ase)enhancer-GAL4* lines that do not contain the repressive element (Fig 2C1-2F1 and 2P), further supporting that the expression of the GAL4 transgenes under the control of the *ase* enhancer fragments that contain the repressive element was reduced due to the feedback inhibition by the misexpressed Tll. The difference of the expression of mCD8-GFP and suppression of Ase is unlikely due to positional effects of these GAL transgenes or differential regulation of the expression of these GAL4 transgenes by endogenous Tll. All these *pDes-(ase)enhancer-GAL4* transgenes were inserted in the same genomic locus and there was no obvious difference in the expression of *UAS-mCD8-GFP* driven by these GAL4 lines when *UAS-Tll* was not co-expressed (Fig 2G–2J and 2Q). Further, no endogenous Tll could be detected in type I NBs in VNCs (Fig 2M–2M") by immunostaining using the existing anti-Tll antibody [22], which could easily detect the endogenous Tll in type II NBs and the optic lobe (Fig 2K–2L"). Therefore, these data provide indirect evidence to support that Tll could bind to the 16-bp repressive element to suppress Ase expression.

## The repressive element contains the consensus Tll binding sequence

Then we examined if the 16-bp repressive element contains a consensus Tll binding sequence 5'-AAGTCA [23]. Indeed, when we searched potential Tll bind sites in the 675-bp region from -1,734bp to -1,060bp using the software FIMO (Find Individual Motif Occurrences) [24], we identified two potential Tll binding motifs that are adjacent to each other with a space of 11 nucleotides in between and have sequences similar to the consensus Tll binding sequence 5'-AAGTCA half-site. One motif (5'-CGTCGTCAAA, named as "Tll_site_L" here) is located from -1,292bp to -1,283bp, which is within the 16-bp repressive element that we mapped. The other (5'-CCGAGTCAAA, named as "Tll_site_R") is located further downstream, from -1,271bp to -1,262bp (Fig 3A).

To examine if Tll_site_R was also required to suppress Ase expression in type II NBs, we generated a *pDes(ase)enhancer-GAL4* reporter driven by the *ase* enhancer fragment from -1,295bp to -1,060bp that contains the wild type Tll_site_L but a mutated Tll_site_R. We

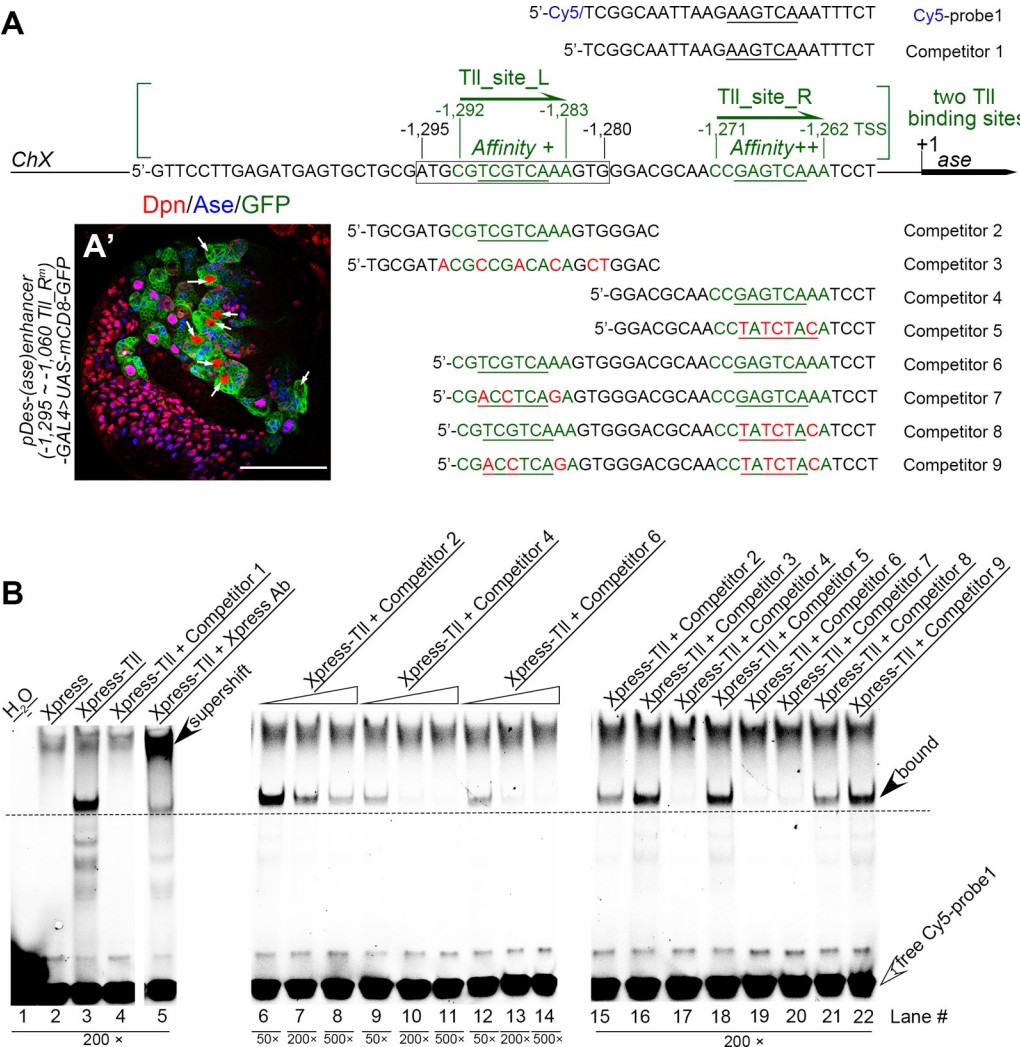

**Fig 3. Tll directly binds to *ase* regulatory regions.** (A) Sequences of a Cy5 labeled probe and various competitors used in EMSAs are shown. Cy5-probe1 and competitor 1 contain a sequence from a *kr* enhancer sub-fragment with the underlined hexameric 5'-AAGTCA half-site core sequence that are bound by Tll proteins. Other competitors contain sequences from the *ase* regulatory region with the predicted Tll binding sites, Tll_site_L and/or Tll_site_R, shown in green. The underlined sequences are the core half-sites of the Tll DNA binding motifs. Red nucleotides indicate mutations introduced into the predicted Tll binding sites. (A') mCD8-GFP driven by the *pDes-(ase)enhancer-GAL4* driver that contains a wild type Tll_site_L site and a mutated Tll_site_R site is expressed in both type I NB lineages and type II NB (white arrows) lineages. Scale bars equal 50μm. (B) Binding between Xpress-tagged Tll proteins and Cy5-probe1 and its competition by indicated competitors. Open arrowheads point to free Cy5-probe1 at the bottom of gels, while filled arrowheads point to the lagged bands of the complexes of Tll-Cy5-probe1 or the super-shifted band of the complexes of Xpress Ab-Tll-Cy5-probe1. Numbers under corresponding lanes indicate the ratios of competitors to probes. Specific binding between Tll and Cy5-probe1 is detected in lane #3 and verified in lane #5 by the presence of a super-shifted band when the Xpress antibody is present. Specific competition is detected when competitors that contain the wild type sequence of either one (competitors #2, #4, #7, and #8, corresponding to lanes #6–11 and #20–21) or both (competitor #6, corresponding to lanes #12–14 and #19) of the predicted Tll binding sites are present, but not when competitors that contains only mutated sites (competitors #3, #5, and #9, corresponding to lanes #16, #18, and #22). However, competitors that contain only the wild type Tll_site_L site (competitors 2 and 8, corresponding to lanes #6–8, #15, and #21) have relatively weak competition compared to competitors that contain only the wild type Tll_site_R site (competitors #4 and #7, corresponding to lanes #9–11 and #20) or both wild type sites (competitor #6, corresponding to lanes #12–14 and #19).

found that this GAL4 reporter was expressed not only in type I NBs but also in all type II NBs (Fig 3A'), indicating that Tll_site_R is also essential for the suppression of Ase in type II NBs.

## Tll binds directly to the predicted Tll binding motifs in the *ase* regulatory region

To determine if Tll binds to the two predicted motifs, we then performed electrophoretic mobility shift assays (EMSAs). This technique detects the binding between a protein of interest and its potential DNA binding motifs by showing a retarded migration of the protein-DNA complex compared to the unbound DNA fragments, which are labelled with fluorophores or radioactive tags and serve as probes. The specificity of the binding could be demonstrated by competition of the binding by competing DNAs (or competitors) that harbor either canonical binding motifs or experimentally verified binding motifs of the protein. We first examined if DNA fragments containing either one or both of these motifs could compete with a 25-bp DNA fragment containing the canonical 5'-AAGTCA Tll-binding core site from the *krüppel (kr)* enhancer, which has been demonstrated to bind strongly to Tll [25]. We expressed Xpress-tagged Tll *in vitro* for EMSAs and the 25-bp *kr* enhancer DNA fragment with the canonical Tll binding core site was labeled with Cy5 at the 5'-end and used as a probe (Cy5-p-robe 1) (Fig 3A). We verified the specific binding between Xpress-Tll protein and Cy5-probe1 (Fig 3B, lane #3) by specific competition by the cold competitor (competitor 1) containing the same sequence as Cy5-probe1 (Fig 3B, lane #4) and the presence of a super-shifted band when anti-Xpress antibody was present (Fig 3B, lane #5). Then we tested competition with DNA fragments that contain either one (competitors 2 and 4) or both (competitor 6) of the predicted Tll binding motifs. Our results showed that all these competitors could successfully compete with Cy5-probe1 in a dose-dependent manner (Fig 3B, lanes #6–14, #15, #17, and #19). However, the DNA fragments that contain the wild type Tll_site_L motif alone (competitor 2) (Fig 3B, lanes #6–8 and #15) could not compete as strongly as DNA fragments that contain the wild type Tll_site_R motif alone (competitor 4) (Fig 3B, lanes #9–11 and #20) or both motifs (competitor 6) (Fig 3B, compare lanes #6–8 to lanes #9–14), suggesting that the Tll_site_L motif has lower affinity with Tll than the Tll_site_R motif.

To further determine the competitions were mediated by the binding motifs, we also used DNA fragments that carried mutations in one (competitors 3, 5, 7, and 8) or both (competitor 9) of the motifs for the competition. In contrast to their wild type counterparts (competitors 2, 4, and 6), DNA fragments containing only the mutated Tll_site_L (competitor 3), or mutated Tll_site_R (competitor 5), or both mutated motifs (competitor 9) failed to compete with the probe (Fig 3B, compare lanes #16, #18, #22 with lanes #15, #17, #19, respectively). The DNA fragment containing a mutated Tll_site_L motif but a wild type Tll_site_R (competitor 7) still could compete for the binding as strongly as the DNA fragment containing both wild type motifs (competitor 6), whereas the competition by the DNA fragment containing a mutated Tll_site_R motif but a wild type Tll_site_L (competitor 8) was compromised although still stronger than the DNA fragment containing both mutated motifs (competitor 9) (Fig 3B, lanes #19–22). These results confirm that the both predicted motifs could bind to Tll, but the Tll_site_R motif has stronger affinity with Tll.

In order to directly visualize the binding of the predicted motifs with Xpress-Tll, we also synthesized Cy5-labeled probes containing the Tll_site_L motif alone (Cy5-probe2), the Tll_site_R motif alone (Cy5-probe3), or both (Cy5-probe4) for EMSAs (S1A Fig). Consistent with the above competition results, we were able to detect binding of Cy5-probes3 and Cy5-probe4 with Xpress-Tll (S1C and S1D Fig, lanes #25 and #37), which could be completely competed by cold competitors (competitors 4 and 6) with the same sequences (S1C and S1D Fig, lanes

#26 and #38) or the competitor 1 that carried the canonical Tll binding motif from *kr* (S1D Fig, lane #42), but not by the competitors that carried mutations in the corresponding motifs (competitors 5 and 9) (S1C and S1D Fig, lanes #27 and #41). Furthermore, we observed that the binding of Xpress-Tll with either Cy5-probe3 or Cy5-probe4 generated two separate bands when we run the gels for a long time (S1E Fig, lane #43 and 44), suggesting the Tll can bind to the motifs either as a monomer or homodimer as reported previously [25–28]. However, we could not detect direct binding of Cy5-probe2 that carried the Tll_site_L motif alone (S1B Fig, lane #23) likely due to its relatively low affinity with Tll. In any event, the DNA fragments that contain the wild type (competitor 2) but not mutated (competitor 3) sequence of the Tll_site_L motif could successfully compete with the Cy5-probe3 for the binding with Xpress-Tll in a dose-dependent manner (S1C Fig, lanes #28–33). The competitor 8 containing a wild type Tll_site_L motif and a mutated Tll_site_R motif could also partially compete with the Cy5-probe4 for the binding (S1D Fig, lane #40), although the competition was not as strong as the competitor 7 that carried a mutated Tll_site_L motif and a wild type Tll_site_R motif (S1D Fig, lane #39).

Taken together, the EMSA results demonstrate that both predicted Tll binding motifs can bind to Tll, but the Tll_site_R motif has higher affinity than the Tll_site_L motif. Our results are consistent with a recent finding that Tll could bind to a 5-kb region upstream of the *ase* TSS, which harbors the two Tll binding motifs we mapped, *in vivo* [21].

## PntP1 functions as a transcriptional activator to indirectly suppress Ase expression in type II NBs

The finding that Tll binds directly to the ase regulatory region to suppress Ase expression led us to hypothesize that PntP1 activates Tll expression in type II NBs. To test this hypothesis, we first wanted to confirm that PntP1 indeed functions as a transcriptional activator rather than a transcriptional repressor in type II NBs.

To this end, we generated artificial chimeric repressor and activator constructs under the control of the UAS promoter. The repressor construct, *UAS-NLS-EnR-Ets*, expresses a fusion protein with the Ets DNA binding domain (aa. 512–597) of PntP1 [29,30] at the C-terminus and the repressor domain of Engrailed (aa. 1–298) (EnR) [31,32] at the N-terminus (S2A Fig). The activator construct, *UAS-NLS-VP16AD-Ets*, expresses a fusion protein with the Ets domain (aa. 512–597) of PntP1 at the C-terminus and the activation domain of VP16 (VP16AD)) [33,34] at the N-terminus (S3A Fig). Both constructs also contain a sequence encoding a PKKKRKV nuclear localization signal (NLS) from the simian virus 40 (SV40) large T antigen (short for NLS) [35] that would be fused to the chimeric activator/repressor proteins at the N-terminus to facilitate nuclear import of the chimeric proteins (S2A and S3A Figs). As controls, we also generated *UAS-pntP1* [10], *UAS-NLS-Ets*, and *UAS-NLS-EnR* constructs that express the wild type PntP1 protein, the Ets domain fused with the NLS, and the EnR fused with the NLS, respectively (S2A Fig). These constructs were then expressed either in type I NBs using *insc-GAL4* [36] as a driver or in type II NBs using *pntP1-GAL4* [10] as a driver to test if they can functionally mimic or antagonize the wild type PntP1 protein, respectively.

Our results showed that the chimeric repressor protein NLS-EnR-Ets could not functionally mimic the wild type PntP1 protein but rather antagonized the activity of endogenous PntP1 proteins. As we showed previously [9,10,14], expression of *UAS-pntP1* driven by *insc-GAL4* suppressed Ase expression in more than 80% of type I NBs and induced generation of Ase[+] Dpn[+] mature INP (mINP)-like cells in over 15% of type I NB lineages in the VNC (S2B-S2C' and S2M Fig). However, expression of *UAS-NLS-EnR-Ets* or other control vectors *UAS-NLS-Ets* or *UAS-NLS-EnR* in type I NBs neither suppressed Ase expression nor induced the

generation of INP-like cells in the VNC (S2D-S2F' and S2M Fig). When *UAS-NLS-EnR-Ets* was expressed in type II NB lineages, we found that Ase was ectopically expressed in over 80% of type II NBs. About 20% of type II NB lineages lost all INPs and were transformed into type I NB-like lineages. Moreover, the number of type II NBs was also increased from 8 to more than 20 per brain lobe (S2G-S2G', S2I-S2I', and S2N Fig). These phenotypes are specific to the chimeric repressor as expression of other control constructs, *UAS-pntP1*, *UAS-NLS-EnR*, or *UAS-NLS-Ets*, did not cause any obvious phenotypes (S2H-S2H', S2J-S2K', and S2N Fig). The phenotypes caused by the expression of the chimeric repressor in type II NBs are similar to those caused by PntP1 knockdown in type II NBs (S2L-S2L' and S2N Fig), indicating that the chimeric repressor NLS-EnR-Ets antagonized the activity of endogenous PntP1 proteins. Therefore, PntP1 unlikely functions as a transcriptional repressor in type II NB lineages.

To determine if PntP1 acts as a transcriptional activator, we then expressed the chimeric activator construct *UAS-NLS-VP16AD-Ets* in type I NBs. To our surprise, expression of *UAS-NLS-VP16AD-Ets* neither suppressed Ase nor promoted the generation of INP-like cells in any type I NB lineages as the expression of *UAS-pntP1* did (S3B-S3D' and S3I Fig). We reasoned that the lack of the phenotypes could be due to the missing of protein-protein interaction domains, which are common for Ets family proteins and essential for recruiting co-activators to activate target gene expression [37,38]. Therefore, we generated other three constructs, *UAS-NLS-pntP1(1/2N)-VP16AD-Ets*, *UAS-NLS-VP16AD-pntP1(1/2N)-Ets*, *UAS-NLS-VP16AD-pntP1(1/2C)-Ets*, which express chimeric activators harboring the Ets domain, the VP16AD, and the N-terminal (pntp1(1/2N) (aa. 1–255) or the C-terminal (pntp1 (1/2C) aa. 256–511) half of PntP1 at either the N-terminus or C-terminus of VP16AD (S3A Fig). Our results showed that expression of *UAS-NLS-VP16AD-pntP1(1/2C)-Ets* led to the suppression of Ase in an average of 45% of type I NBs and induction of Ase$^+$ Dpn$^+$ mINP-like cells in over 6% of type I NB lineages in the VNC (S3G-S3G' and S3I Fig), whereas expression of either *UAS-NLS-pntP1(1/2N)-VP16AD-Ets* or *UAS-NLS-VP16AD-pntP1(1/2N)-Ets* did not (S3E-S3F' and S3I Fig). To ensure that the effects of *NLS-VP16AD-pntP1(1/2C)-Ets* were not due to inclusion of endogenous activation domain of PntP1 in the pntP1(1/2C) fragment, we also generated a control construct *UAS-NLS-pntP1(1/2C)-Ets*, which does not express the VP16AD. However, expressing *UAS-NLS-pntP1(1/2C)-Ets* neither suppressed Ase expression in type I NBs nor induced any INP-like cells (S3H-S3I Fig), indicating that the pntP1(1/2C) fragment does not contain the endogenous activation domain. Taken together, the results from these artificial chimeric activator proteins (summarized in S3J Fig) support that PntP1 functions as a transcriptional activator to indirectly suppress Ase expression in type II NB lineages, but its transcriptional activation activity not only relies on its Ets domain and activation domain, but also requires other yet-to-be-characterized functional domain(s) potentially involved in protein-protein interactions in the C-terminal half of the protein (S3K Fig).

## PntP1 is both necessary and sufficient for the activation of Tll expression

We then tested whether Tll is a downstream target of PntP1 in type II NB lineages. We first examined if PntP1 was necessary for Tll expression in type II NBs. We knocked down PntP1 in type II NBs and examined Tll expression using a *tll-EGFP* reporter line, which carries an EGFP coding sequence in the *tll* locus and expresses Tll-EGFP fusion proteins [21,39]. Tll-EGFP is robustly expressed in wild type type II NBs (Fig 4A–4A" and 4I). However, when PntP1 was knocked down in type II NBs, Tll-EGFP expression was drastically reduced by 80% on average and Ase was ectopically turned on in an average of 90% of type II NBs (Fig 4B–4B" and 4I–4J). Restoring Tll levels by expressing *UAS-Tll* in PntP1 knockdown type II NBs led to the suppression of Ase (Figs 4C–4D", 4J and S4A–S4E). Furthermore, co-expression of *UAS-*

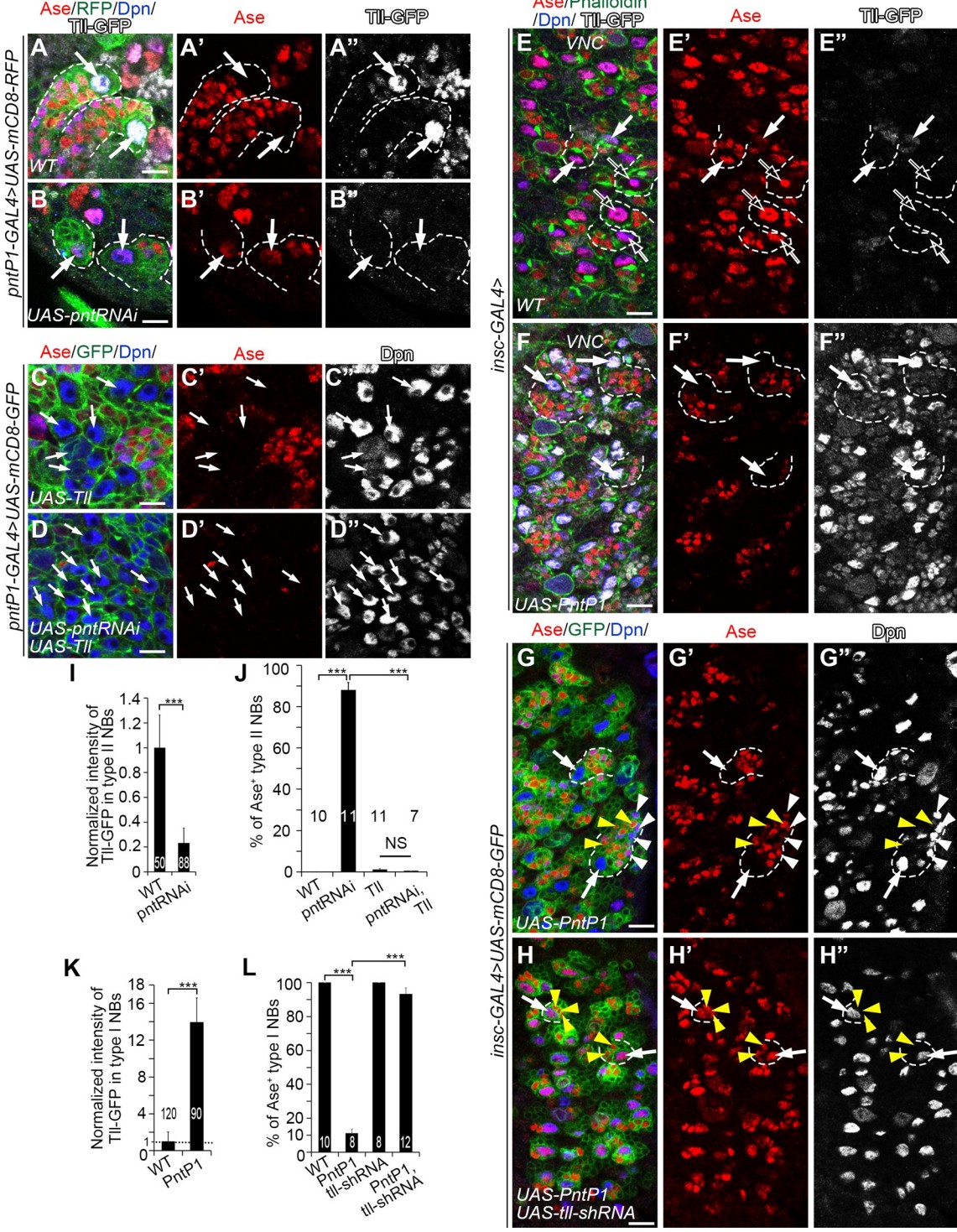

**Fig 4. Tll is the PntP1 downstream effector that represses Ase expression in type II NBs.** In all images, white arrows point to some representative type I NBs or type II NBs. Dashed lines outline some type I NB or type II NB lineages. White arrowheads point to some Ase⁺ Dpn⁺ mINP-like cells and yellow arrowheads point to some Ase⁺ Dpn⁻ GMCs as examples. Type II NB lineages in the brain are labeled with mCD8-RFP or mCD8-GFP (in green) driven by *pntP1-GAL4* and type I NB lineages in the VNC are labeled with Phalloidin or mCD8-GFP (in green) driven by *insc-GAL4*. The brains/VNCs are counterstained with anti-Ase (in red), anti-Dpn (in blue) and/or anti-GFP (in white) antibodies. Scale bars equal 10μm. (A-B") Tll is specifically expressed in wild type type II NBs, which are Ase⁻ (A-A"), while knockdown of PntP1 leads to the abolishment of Tll expression and ectopic Ase expression in type II NBs (B-B"). (C-D") Expressing *UAS-Tll* in type II NBs results in generation of supernumerary type II NBs, which are Ase⁻ (C-C"), and

knockdown of PntP1 by expressing *UAS-pnt RNAi* does not lead to ectopic Ase expression in type II NBs when *UAS-Tll* is expressed simultaneously (D-D"). (E-F") Wild type type I NBs in the VNC are Ase positive and have no (open arrows) or low levels (white arrows) of Tll-GFP expression at late 3rd instar larval stages (E-E"), while misexpressing *UAS-PntP1* activates ectopic Tll-GFP expression in all type I NBs and represses Ase expression (F-F"). (G-H") Misexpressing *UAS-PntP1* suppresses Ase expression in type I NBs in the VNC and induces the generation of Ase+ Dpn+ mINP-like cells (G-G"), whereas simultaneous knockdown of Tll prevents the suppression of Ase and induction of mINP-like cells by PntP1 misexpression (H-H"). (I-J) Quantifications of Tll-GFP intensity in wild type or PntP1 knockdown type II NBs (I), and the percentage of Ase+ type II NBs (J) in the brains with indicated genotypes. Values of the bars are mean ± SD. ***, $P < 0.001$. NS, not significant. (K-L) Quantifications of Tll-GFP intensity in wild type and PntP1 misexpressing type I NBs (K), and the percentage of Ase+ type I NBs in the VNCs with indicated genotypes (L). Values of the bars are mean ± SD. ***, $P < 0.001$.

*pnt RNAi* and *UAS-Tll* induced more supernumerary type II NBs than the expression of either one of them (Figs 4B–4D" and S4B–S4E) probably because both knockdown of PntP1 and overexpression of Tll would promote dedifferentiation of imINPs into type II NBs (S4A–S4D and S4F Fig) as previously reported [21,40,41]. These results demonstrate that PntP1 is necessary for Tll expression in type II NBs and Tll acts downstream of PntP1 to repress Ase expression in type II NBs.

Next, we examined if PntP1 is sufficient to activate Tll expression. We misexpressed *UAS-PntP1* in type I NBs using *insc-GAL4* as a driver and examined the expression of Tll-EGFP or endogenous wild type Tll protein. In wild type animals, Tll-EGFP is weakly expressed in a small subset of type I NBs in the VNC (Fig 4E–4E" and 4K), whereas the endogenous wild type Tll proteins are not detectable using the anti-Tll antibody (Figs 2M–2M" and S5A–S5A"). When PntP1 was misexpressed in type I NBs, it dramatically increased Tll-EGFP expression by about 14-fold in all type I NBs in the VNC, and the expression of Tll-EGFP protein also weakly persisted in their progeny (Fig 4F–4F" and 4K). Similarly, the expression of endogenous Tll was also activated in type I NBs by PntP1 misexpression (S5B–S5B" Fig). Consistently, Ase expression was abolished in about 90% of type I NBs in the VNC (Figs 4F–4F", 4K–4L, and S5B–S5B"). In order to verify that the loss of Ase expression was due to activation/increase in Tll expression, we simultaneously knocked down Tll while misexpressing PntP1 in type I NBs. We found that Ase expression was restored in 93% of type I NBs (Fig 4G–4H" and 4L). These results suggest that misexpressing PntP1 is sufficient to activate the expression of Tll, which in turn suppresses Ase expression, in type I NBs.

## PntP1 binds directly to the enhancer region of *tll*

Next, we wanted to determine if PntP1 transactivates Tll by directly binding to *tll* enhancer regions. A previous study [42] has shown that a 3.1-kb enhancer fragment, *R31F04*, located at -4,912bp ~ -1,855bp upstream of the *tll* transcription start site, can drive the expression of the *GAL4* transgene specifically in type II NBs, whereas another overlapping fragment *R31D09*, located at -2,893bp ~ -157bp, cannot (Fig 5A and 5B), suggesting that the enhancer region from -4,912bp to -2,893bp contains *cis*-elements required for Tll expression in type II NBs. We used the software FIMO to search PntP1 binding sites in this 2-kb enhancer region and found seven putative PntP1 binding sites (sites #1–7) containing the consensus 5'-GGAA/T core sequence (Fig 5A).

To determine if PntP1 binds to these putative binding sites, we performed EMSAs and Chromatin Immunoprecipitation (ChIP) in combination with quantitative PCR (qPCR). For EMSAs, we used a previously confirmed DNA fragment containing the sequence bound by PntP1 from the *erm* enhancer *R9D11* [16] as a probe (Cy5-probe5) and the DNA fragments containing wild type or mutated sequences of the individual predicted PntP1 binding sites as competitors (competitors T1-T7 and competitors T1Mu-T7Mu) (Fig 5C). We confirmed the binding of the probe with PntP1 by the presence of a band of the Xpress-PntP1-probe

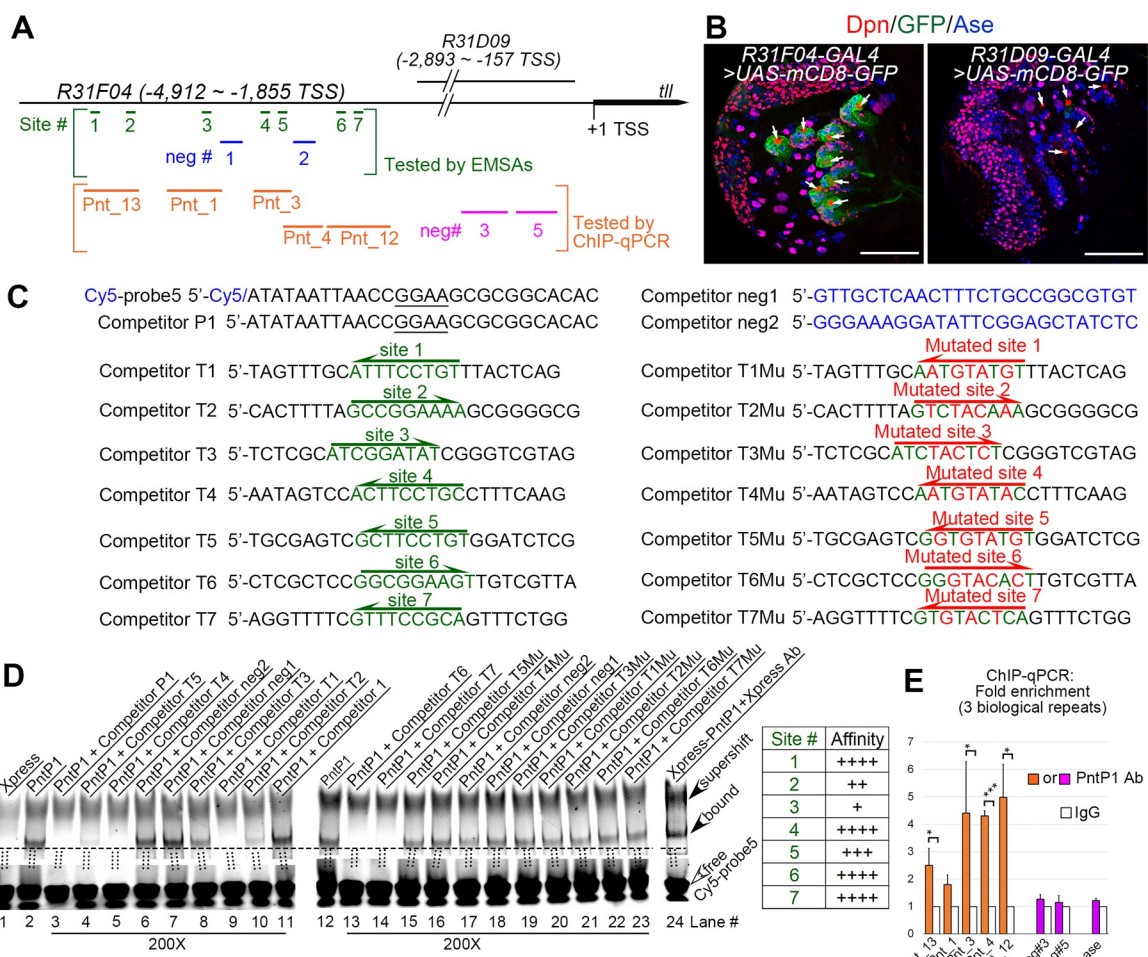

**Fig 5. PntP1 directly binds to the *tll* enhancer in type II NBs.** (A) A diagram of locations of putative PntP1 binding sites or negative control sites in the *tll* enhancer regions of *R31F04* or *R31D09*. Green lines and blue lines within square brackets indicate the PntP1 binding sites (sites #1–7) and the sequences (neg#1–2) lacking PntP1 binding sites, respectively, that are tested by EMSAs. Orange lines and magenta lines within the orange square brackets indicates DNA fragments with (Pnt_13, Pnt_1, Pnt_3, Pnt_4, Pnt_12) or without (neg#3, neg#5) corresponding PntP1 binding sites, respectively, that are tested by ChIP-qPCR. (B) GAL4 under the control of the *tll* enhancer subfragment *R31F04* but not *R31D09* specifically activates mCD8-GFP expression in type II NB lineages. White arrows point to type II NBs. Scale bars equal 50μm. (C) Sequences of competitors and probes used for EMSAs. Nucleotides in green indicate sequences of putative PntP1 binding motifs and nucleotides in red indicated mutations introduced into the putative PntP1 binding motifs. Underlined nucleotides in the Cy5-probe5 and the competitor P1 indicates the 5'-GGAA core sequence of the PntP1 binding motif. The direction of arrows indicates the orientations of the consensus 5'-GGAA/T-3' core sequences. (D) EMSAs of the binding of Xpress-tagged PntP1 with Cy5-probe5 that contains a PntP1 binding motif from the *erm* enhancer fragment *R9D11* and its competition by indicated competitors that contains corresponding wild type or mutated putative PntP1 binding motifs. Specific binding between Xpress-PntP1 and Cy5-probe5 is detected in lanes #2 and #12 and is verified by the presence of a super-shifted band in the presence of the Xpress antibody (lane #24) and the competition by the cold probe (competitor P1) (lane #3). Competitors that contain the wild type PntP1 binding sites (competitors T1-T7, corresponding to lanes #4–5, #8–10, and #13–14) can all compete with the Cy5-probe5 for the binding with Xpress-PntP1 to a various of degrees, but the competitors that contain mutated PntP1 binding motifs (competitors T1Mu-T7Mu, corresponding to lanes #15–16, and #19–23) or no PntP1 binding motifs (competitors neg1 and neg2 or the competitor 1 that contains the sequence from the *kr* enhancer, corresponding to lanes #6–7, #11, and #17–18) cannot. The relative binding affinity of the seven putative PntP1 binding site is summarized in the table to the right. Open arrowheads point to free probes while filled arrowheads point to the bands of PntP1-Cy5-probe5 complexes or the super-shifted band of the Xpress Ab-PntP1-Cy5-probe5 complex. The ratio of the competitor to the probe is 200 for all competitors tested. (E) DNA fragments (Pnt_13, Pnt_1, Pnt_3, Pnt_4, Pnt_12) that contain corresponding predicted PntP1 binding sites are all pulled down by the anti-PntP1 antibody from larval brains in the ChIP-qPCR assays, but not the DNA fragments neg#3 or neg#5 from the *R31D09* enhancer region or a DNA fragment from the *ase* enhancer that do not contain putative PntP1 binding sites. The quantification data represent the Mean ± SD of the fold changes normalized to IgG controls of three biological replicates. *, $P < 0.05$. ***, $P < 0.001$.

complex, super-shift of the band in the presence of the anti-Xpress antibody, and loss of the band in the presence of a competitor containing the same sequence as the probe (competitor P1) (Fig 5C and 5D, lanes #1–3 and 24). Our results showed that competitors T1-T7 containing the wild type sequences of the predicted PntP1 binding sites all could compete with the probe for binding to Xpress-PntP1 to various degrees (Fig 5C and 5D, lane #4–5, 8–10, and 13–14), but the competitors containing the mutated sequences of the predicted PntP1 binding sites (competitors T1Mu–T7Mu) or no predicted PntP1 binding sites (competitors neg1 and neg2 and the competitor 1 from the *kr* regulatory region) could not (lane #6–7, 11, and 15–23). Based on the extent of competition, the affinity of the predicted binding sites for PntP1 is in the following order from high to low: site #1, site #4, site #6, site #7 > site # 5> site #2 > site #3. Similar EMSA results were also obtained when a DNA fragment with the exact same sequence as competitor T4, which contains the predicted PntP1 binding site 4, was used as a probe (Cy5-probe 6) (S6A Fig).

In order to determine if PntP1 directly binds the *tll* enhancer region *in vivo*, we performed ChIP assays using Brat knockdown larval brains that were enriched with type II NBs followed by qPCR using 5 distinct sets of primers flanking one or two of the predicted PntP1 binding sites (Fig 5A). Previous studies have shown that type II NB-enriched *brat* mutant larval brains could be used for determining DNA binding sites of the chromatin modifying protein Trithorax and chromatin methylation/acetylation status of particular enhancer regions of type II NB lineage-specific gene *erm* by ChIP-qPCR [16,43]. Consistent with the EMSA results, our ChIP-qPCR results showed that the anti-PntP1 antibody could pull down DNAs from the *tll* enhancer region containing any one of the predicted binding sites, but not DNAs from the *tll* enhancer regions (neg #3 and neg #5) that do not contain the predicted PntP1 binding sites or from *ase* enhancer regions, the latter of which provides additional evidence to support that PntP1 doesn't suppress *ase* expression directly (Fig 5E). These ChIP-qPCR results demonstrate that endogenous PntP1 proteins bind to the *tll* enhancer region directly.

## Tll is required to maintain PntP1 expression in type II NBs

Though we have shown that Tll is activated by PntP1, there is evidence showing that ectopic expression of Tll in type I NBs also induces PntP1 expression [21], leading us to hypothesize that Tll may provide positive feedback to maintain PntP1 expression in type II NBs. To test this hypothesis, we examined PntP1 expression in Tll knockdown type II NBs by examining the expression of *UAS-mCD8-GFP* driven by *pntP1-GAL4* or the expression of endogenous PntP1 proteins. Since the *pntP1-GAL4* is an enhancer trap line, in which the *GAL4* transgene is inserted at 347 bps upstream of the TSS of *pntP1* [10], the levels of mCD8-GFP driven by *pntP1-GAL4* should reflect the activity of the *pntP1* enhancer/promoter and endogenous PntP1 levels in type II NB lineages. We found that when Tll was knocked down, the expression of mCD8-GFP in type II NBs were significantly reduced by around 75% (Fig 6A–6B'''). Our immunostaining results also showed that endogenous PntP1 expression was almost abolished in Tll knockdown type II NB lineages (Fig 6C–6D" and 6H). Consistent with the reduction of PntP1 expression, type II NB lineages were transformed into type I-like NB lineages as indicated by ectopic Ase expression in the NBs (Fig 6B–6B" and 6I), loss of Ase+ imINPs and mINPs, and loss of the expression of *R9D11-mCD8-GFP*, which reflects endogenous Erm expression in imINPs (Fig 6E–6G'). Therefore, Tll is required for maintaining the expression of PntP1 and there is a positive feedback loop between PntP1 and Tll in type II NBs.

The positive feedback regulation of PntP1 by Tll led us to ask whether in addition to directly suppressing Ase expression, Tll could also be indirectly regulating the expression of other unknown downstream targets of PntP1, which may act in parallel to Tll to suppress Ase. To

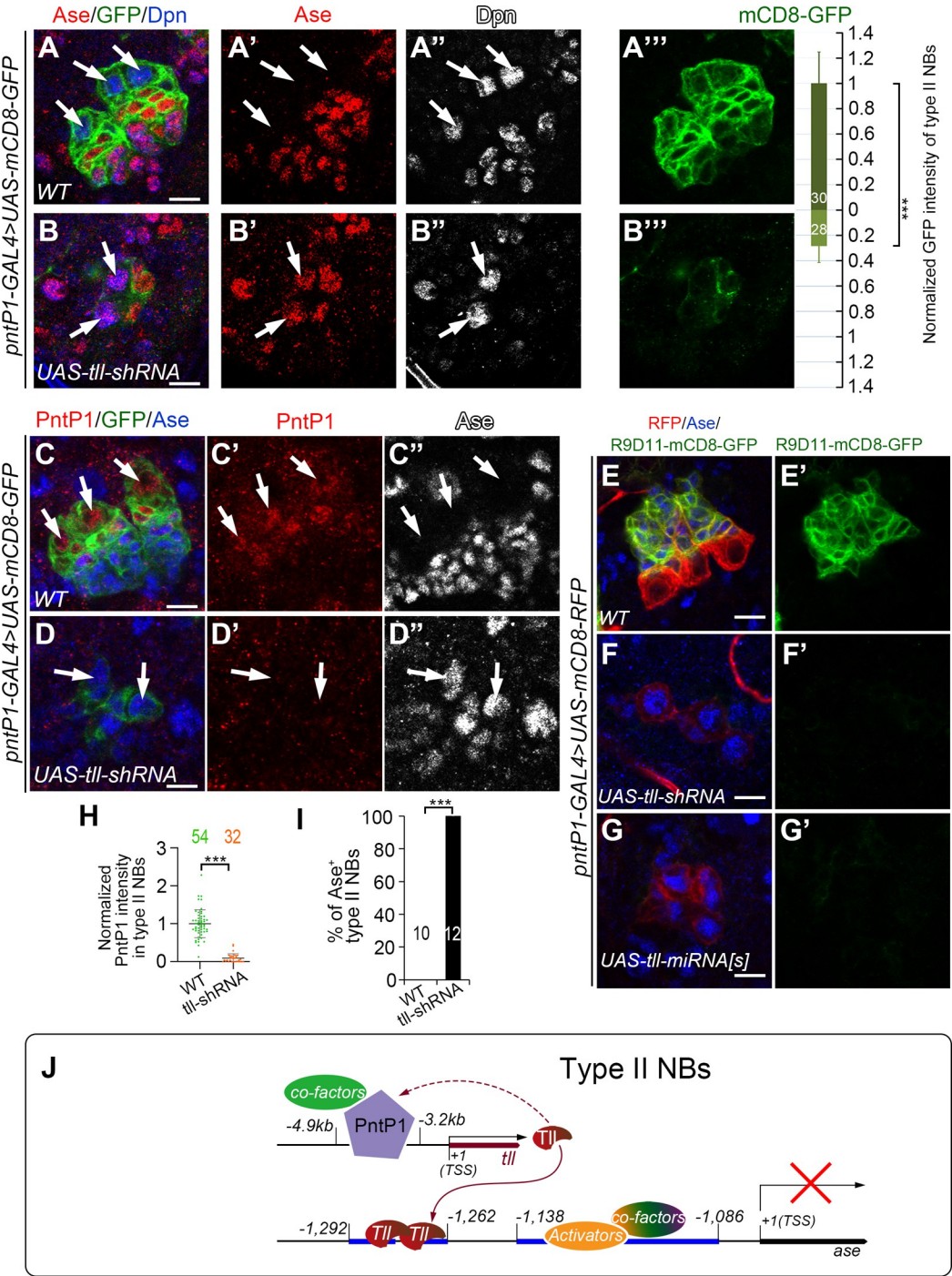

**Fig 6. PntP1 is maintained by Tll in type II NBs.** Type II NB lineages are labeled with mCD8-GFP/RFP driven by *pntP1-GAL4* and counterstained with anti-Ase, anti-Dpn, and/or anti-PntP1 antibodies. White arrows point to type II NBs. Scale bars equal 10μm. (A-B''') Wild type type II NB lineages have no Ase expression in the NBs and have higher expression levels of mCD8-GFP driven by *pntP1-GAL4* (A-A'''), whereas knockdown of Tll in type II NBs leads to ectopic Ase expression in type II NBs and a decrease in mCD8-GFP expression by 75% (B-B'''). mCD8-GFP intensity in the wild type type II NBs is normalized as 1 (A and A'''). Values of the bars are mean ± SD. Numbers within the bars represent sample size. ***, $P < 0.001$. (C-D'') Wild type type II NBs express PntP1 but not Ase (C-C''), while knockdown of Tll leads to loss of PntP1 expression and ectopic Ase expression in type II NBs (D-D''). (E-G') The Erm reporter *R9D11-mCD8-GFP* is expressed in Ase[+] imINPs in wild type type II NB lineages (E-E'), but is abolished when Tll is knocked down in type II NBs by expressing *UAS-tll-shRNA* (F-F') or *UAS-tll-miRNA[S]* (G-G'). (H-I) Quantifications of PntP1 intensity in type II NBs (H) and the percentage of Ase[+] type II NBs (I) in wild type and Tll knockdown brains. Values represent mean ± SD. The

numbers on the graphs are the number of type II NBs (H) or brains lobes (I) examined. ***, P<0.001. NS, not significant. (J) A working model shows that PntP1 is specifically expressed in type II NBs and directly activates Tll expression by binding to the *tll* enhancer region. Tll binds two hexameric half-sites in the *ase* enhancer to suppress its transcription. Meanwhile, Tll maintains the expression of PntP1 through an unknown pathway.

test this possibility, we tried to restore PntP1 levels in Tll knockdown type II NBs by expressing *UAS-PntP1* and checked if the ectopic Ase expression would be inhibited. When *pntP1-GAL4* drove the expression of *UAS-PntP1* in wild type animals, PntP1 levels in type II NBs were elevated by 2.2 folds without disrupting type II NB lineage development (S7A–S7A" and S7J Fig). However, to our surprise, expression of *UAS-PntP1* driven by *pntP1-GAL4* failed to restore PntP1 protein levels in 76.8% of Tll knockdown type II NBs and Ase is still ectopically expressed (S7B-S7B" and S7J–S7K Fig). We speculated that it was probably because that *pntP1-GAL4* expression was decreased, as reflected by reduced expression of mCD8-GFP (Fig 6B"'), due to the loss of the positive feedback regulation by Tll. Nevertheless, PntP1 expression was restored to levels comparable to those in wild type type II NBs in the rest 23.2% of Tll knockdown type II NBs, yet Ase was still ectopically expressed in these NBs (S7C–S7C" and S7J–S7K Fig). In order to more consistently restore PntP1 expression in Tll knockdown type II NBs, we then used *R31F04-GAL4* to drive the expression of *UAS-PntP1* while knocking down Tll in type II NBs. Since PntP1 activates *tll* expression through binding to this *R31F04* enhancer region, expression of *UAS-PntP1* driven by *R31F04-GAL4* could potentially exert a positive feedback activation of *R31F04-GAL4* and thus more robust restoration of PntP1 expression in Tll knockdown type II NBs. Indeed, expression of *UAS-PntP1* driven by *R31F04-GAL4* restored PntP1 protein levels in 95.8% of Tll knockdown type II NBs. However, all type II NBs still showed strong Ase expression (S7D–S7E" and S7L–S7M Fig). Therefore, in the absence of Tll, PntP1 can no longer suppress Ase expression in type II NBs. Consistently, we found that Tll misexpression could suppress Ase expression without inducing ectopic PntP1 expression in the majority of type I NBs (S7F–S7G", $S7G_1$–$S7G_1$", and S7N Fig) and that simultaneous knockdown of PntP1 did not restore Ase expression in Tll misexpressing type I NBs (S7H–S7I" and S7N Fig). Therefore, Tll is the major downstream target of PntP1 that is responsible for the suppression of Ase. Other downstream targets of PntP1 may not have significant roles in the suppression of Ase or just play a supplementary role by coordinating with Tll.

## Discussion

In this study, we dissected the molecular mechanism of how PntP1 suppresses Ase expression to specify the type II NB identity. We demonstrate that PntP1 acts as a transcriptional activator to indirectly suppress Ase expression by activating Tll, whereas Tll provides positive feedback to maintain the expression of PntP1 and the type II NB identity (Fig 6J). We further mapped the *cis*-elements that mediate the suppression of Ase by Tll and the activation of Tll by PntP1. Thus, our work reveals mechanistic details of PntP1-mediated suppression of Ase expression and specification of type II NBs and identifies a novel direct target of PntP1 in type II NBs.

### PntP1 functions as a transcriptional activator in type II NB lineages

In this study, we have demonstrated that PntP1 functions as a transcriptional activator by showing that the artificial chimeric repressor protein EnR-Ets antagonizes the function of endogenous PntP1 proteins when expressed in type II NB lineages and that the artificial chimeric activator proteins VP16AD-pntP1(1/2C)-Ets could functionally mimic endogenous PntP1 protein when expressed in type I NBs. Our results are in line with a previous *in vitro*

study showing that PntP1 substantially activates bacterial chloramphenicol acetyltransferase (CAT) reporter expression under the control of Ets binding sites [30]. Interestingly, our results show that the chimeric protein VP16AD-Ets is not sufficient to functionally mimic endogenous PntP1 proteins even though the Ets domain can bind to PntP1 target DNAs as demonstrated by the antagonization of PntP1's function by the EnR-Ets protein. Only when the C-terminal sub-fragment (aa. 256–511) of PntP1 is included can the chimeric activator protein functionally mimic endogenous PntP1 proteins. The Ets family proteins usually recruit additional cofactors to activate/repress target gene expression. We think that this C-terminal fragment likely contains protein-protein interaction domains that are essential to recruit its cofactors. Since neither the pntP1(1/2C)-Ets truncated protein nor the chimeric protein VP16AD-pntP1(1/2N)-Ets is able to mimic wild type PntP1 protein's function, it is unlikely that this C-terminal sub-fragment contains the activation domain of PntP1 and the other N-terminal sub-fragment [pntP1(1/2N)] contains the protein-protein interaction domains instead. Otherwise, with the VP16AD functioning as an activation domain and the N-terminal sub-fragment recruiting cofactors, the chimeric VP16AD-pntP1(1/2N)-Ets protein would be able to functionally mimic wild type PntP1 protein. Given that a large number of Ets family transcription factors share highly conserved Ets domains but have diverse functions and activities in distinct cell types, the C-terminal sub-fragment of PntP1 may recruit cell-type-specific cofactors to regulate the expression of specific target genes in type II NB lineages, such as Erm [10,16] and Tll, as it has been proposed as a general strategy for Ets family proteins to regulate the expression of tissue-specific target genes [38,44]. However, the regions of PntP1 subfragments that we chose to be included in our chimeric repressor/activator constructs are arbitrarily defined. Our results do not tell the precise boundaries of the activation domains or potential protein-protein interaction domains. To precisely map these functional domains will require more detailed and systemic analyses in the future.

## Tll is a novel direct target of PntP1 that mediates the suppression of Ase

PntP1 performs diverse functions in different cell types in type II NB lineages. For example, in type II NBs, PntP1 is required to suppress Ase expression. In newly generated imINPs, PntP1 prevents premature differentiation of INPs, whereas late during imINP development, it promotes INP cell fate commitment and prevents dedifferentiation of imINPs [10,40]. Therefore, as a transcriptional activator, PntP1 must activate the expression of many different target genes in type II NB lineages. However, Erm is the only direct target that has been identified previously [16]. Here we identify Tll as another direct target of PntP1 that functions primarily in type II NBs to suppress Ase expression. We demonstrate that PntP1 is both necessary and sufficient for Tll expression. We further identify 7 putative binding sites and demonstrate by EMSAs and ChIP-qPCR assays that all these sites can bind to PntP1 both *in vitro* and *in vivo*. Although our ChIP-qPCR was done using type II NB-enriched DNAs isolated from Brat knockdown larval brains, it is unlikely the results are artifacts. A previous study has demonstrated that type II NB-enriched chromatin isolated from *brat* mutant larval brains maintains similar transcriptional status for multiple genes examined, including *pntP1*, as in wild type type II NBs [16]. Furthermore, Brat mainly functions as an RNA-binding protein in the imINPs to promote degradation of mRNAs of self-renewing genes such as *dpn* and *klu* [45–47]. Therefore, loss of Brat unlikely affects the binding of PntP1 to its target DNAs in type II NBs, but we do not fully exclude this possibility particularly because the supernumerary type II NBs in Brat knockdown brains are derived from dedifferentiation of imINPs and it has never been extensively evaluated whether the dedifferentiated type II NBs have the exact same gene expression profiles as wild type type II NBs. It might be helpful to further verify the results by

single-cell ChIP using isolated wild type type II NBs. However, since our bioinformatic prediction of PntP1 binding sites was limited to the enhancer region *R31F04*, we cannot be sure whether any additional PntP1 binding sites exist outside this enhancer region and contribute the activation of *tll* by PntP1. In any event, the seven binding sites we identified within this enhancer region are sufficient for *tll* to be activated by PntP1 in type II NBs as demonstrated by the specific expression of *R31F04-GAL4* in type II NBs.

Our work further demonstrates that Tll is the direct target of PntP1 that mediates the suppression of Ase in type II NBs by showing that simultaneous knockdown of Tll essentially blocks the suppression of Ase by misexpressed PntP1 in type I NBs. By fine mapping the *cis*-repressive elements in *ase* regulatory regions, we identified two Tll binding sites located at -1,292bp ~ - 1,262bp upstream of the *ase* TSS, which is consistent with a recent study showing that Tll binds to a 5-kb region upstream of the *ase* TSS [21]. Although the core hexameric sequences of these two binding sites are not exactly the same as the typical Tll binding hexamer 5'-AAGTCA, our EMSA results demonstrate that Tll can indeed bind to these sites either as a monomer or a homodimer, the latter of which is common for orphan nuclear receptors [48].

In addition to PntP1 binding sites, other studies [41,49,50] report that Suppressor of Hairless [Su(H)], a binding partner of the intracellular domain of Notch, and Zelda (Zld) also bind to the enhancer region of *tll*, implicating that Notch and Zld could be upstream activator of Tll. However, Su(H) and Zld are not just expressed in type II NBs but also in type I NBs [45,49]. Thus, Su(H) and Zld are unlikely to be sufficient to activate Tll, but whether PntP1 acts together with Su(H) and Zld to activate Tll in type II NBs is worth further investigation.

Our previous study shows that PntP1 is expressed not only in type II NBs but also strongly in imINPs [9,10]. However, Tll is primarily expressed in type II NBs but only very weakly in imINPs, and ectopic expression of Tll in imINPs reverts imINPs to type II NBs [21,41]. Therefore, there must be a mechanism to inhibit the activation of Tll by PntP1 in imINPs. A recent study proposed that Erm and Hamlet (Ham) function sequentially to suppresses Tll expression in imINPs based on 1) decreasing or increasing the copy number of the *tll* gene suppresses or enhances the supernumerary type II NB phenotype in *ham erm* double heterozygous mutants, respectively; and 2) overexpressing Erm or Ham in type II NBs inhibits the expression of Tll in type II NBs [41]. However, these data could be also explained by changes in the expression of PntP1 as we demonstrated here that there is a positive feedback loop between PntP1 and Tll. For example, the reduction in the Tll expression resulting from Erm overexpression in type II NBs could be due to inhibition of PntP1 by Erm in type II NB as we reported previously [14,51] rather than direct inhibition of Tll by Erm. Furthermore, Erm and Ham are not expressed in the newly generated imINPs, in which Tll expression is already largely suppressed. Therefore, the suppression of Tll in the newly generated imINPs cannot be explained by the inhibition by Erm and Ham. Other mechanisms are likely involved in suppressing Tll in imINPs.

## A positive feedback loop between PntP1 and Tll

Our results not only demonstrate that PntP1 is a direct upstream activator of Tll but also show that Tll is required to maintain PntP1 expression in type II NBs. Therefore, there is a positive feedback loop between PntP1 and Tll that is essential for maintaining the type II NB identity. Considering that PntP1 misexpression induces Tll expression in all type I NBs and generation of mINPs in a subset of type I NB lineages, whereas Tll misexpression induces PntP1 expression only in a small subset of type I NB lineages and does not induce the generation of mINPs, we think that PntP1 functions as a master regulator of type II NB lineage development and acts upstream of Tll, which in turn suppresses Ase expression in type II NBs. Since Tll mainly

functions as a transcriptional repressor [21,52] as also demonstrated in this study, it is unlikely that Tll directly activates PntP1 expression. Our previous studies suggest that there might be an unknown feedback signal from INPs that could be required for maintaining PntP1 expression [9,51]. Thus, one potential explanation for the loss of PntP1 expression in Tll knockdown type II NBs could be the loss of INPs and their feedback signal resulting from the ectopic Ase expression in type II NBs. However, since Tll misexpression is able to induce PntP1 expression albeit only in a small subset of type I NBs, it is more likely that Tll suppresses the expression of another unknown transcriptional repressor that is normally suppressed by Tll in type II NBs. When Tll is knocked down, this unknown transcriptional repressor could be turned on in type II NBs to suppress PntP1 expression. Whereas in type I NBs, this unknown repressor may be normally expressed to suppress PntP1 expression and misexpression of Tll may relieve the suppression of PntP1 expression.

### Activation of Ase in type I NBs

Our GAL4 reporter assays also identified in the *ase* enhancer region a ~50-bp fragment that is sufficient for activating Ase expression in both type I and type II NBs. Earlier studies show that the Achaete-Scute (AS-C) complex proteins together with Daughterlesss (Da) activate Ase expression in NBs during the initial specification of NBs at embryonic stages by directly binding to four E-boxes in the 5'-UTR of *ase* [18,53]. But how Ase is maintained in NBs once they are specified is not known. Our study identified a distinct enhancer region for activating/maintaining Ase expression in NBs, suggesting that factor(s) other than the AS-C proteins may be involved in activating/maintaining Ase expression in NBs after they are specified. Therefore, the lack of Ase expression in type II NBs is not because of the absence of an activation mechanism, but rather this activation mechanism is actively suppressed by Tll. Identifying the transcriptional activator(s) involved in activating/maintaining Ase expression after NBs are specified may shed a new light on the mechanisms regulating the development and maintenance of type I and type II NBs.

## Material and methods

### Fly stocks

*UAS-mCD8-GFP* or *UAS-mCD8-RFP* lines were used for visualizing type I NB or type II NB lineages in combination with appropriate GAL4 lines, including *pntP1-GAL4* (i.e., *GAL4[14-94]*) [10], *insc-GAL4* [36], *ase(F:2.0)-GAL4* [18,54], *GMR20B05-GAL4* (#49843, Bloomington Drosophila Stock Center (BDSC)), *GMR31F04-GAL4* (#46187, BDSC) and *GMR31D09-GAL4* (#49676, BDSC) [42]. *R9D11-mCD8-GFP* line [10] was used as *erm* reporter and to label INPs. *tll-EGFP* line (#30874, BDSC) was used to examine the expression of Tll. For loss-of-function analyses of Tll or PntP1 by RNAi knockdown, *UAS-pnt RNAi* (#35038, BDSC), *UAS-tll-shRNA* (#330031, Vienna Drosophila Resource Center (VDRC), a gift from Dr. A. H. Brand), *UAS-tll-miRNA[s]* [55] (a gift from Dr. A. H. Brand) were used. *UAS-Tll* (#109680, KYOTO Stock Center, a gift from Dr. A. H. Brand) or *UAS-PntP1* [10] were used to mis/over-express or restore Tll or PntP1, respectively. Other lines including *UAS-HA-PntP1* (a gift form Dr. C. Y. Lee) [43] and *UAS-Brat RNAi* (#34646, BDSC) were used to expand the pool of type II NBs for ChIP assays.

### UAS-transgene expression

For expressing the chimeric activators/repressors or corresponding control constructs, embryos were collected for 8~10 hrs at 25˚C and shifted to 30˚C right away to boost the

efficiency till late 3$^{rd}$ instar stage for phenotypic analyses. For testing or mapping the activity of *ase* enhancer fragments, *pDes-(ase)enhancer-GAL4* lines were crossed with *UAS-mCD8-GFP* lines and embryos were collected for 8~10 hrs and raised at 25˚C till late 3$^{rd}$ instar stage for phenotypic analyses. For knocking down PntP1 and/or restoring Tll expression in type II NBs, embryos were collected for 8~10 hrs at 25˚C and shifted to 30˚C till late 3$^{rd}$ instar stage for phenotypic analyses. For knocking down Tll and/or restoring PntP1 in type II NBs, embryos were collected for 8~10 hrs at 25˚C, raised at the same temperature for additional 2 days and then shifted to 30˚C for additional 6 hrs before dissection. For misexpressing PntP1 or Tll, and/or knocking down Tll or PntP1 in type I NBs, temperature sensitive *tub-Gal80$^{ts}$* (#7017, BDSC) was used in combination with *insc-GAL4* to avoid lethality at embryonic stages. Embryos were collected for 8~10 hrs and raised to hatch at 18˚C. The hatch larvae were then maintained at 30˚C for 3~4 days to maximize the efficiency. When *R31F04-GAL4* was used to express *UAS-tll-shRNA* and/or *UAS-PntP1*, embryos were collected for around 12 hrs at 25˚C, raised at the same temperature for additional 2~3 days before dissection. For phenotypic analyses of the misexpression of *UAS-Tll* in type I NBs driven by distinct *pDes-(ase)enhancer-GAL4* drivers, embryos were collected for 8~10 hrs at 25˚C and shifted to 30˚C right away till 3$^{rd}$ instar stage.

## Immunostaining and confocal microscopy

*Drosophila* larvae at desired developmental stages were sacrificed and dissected. Brain lobes along with VNCs were immunostained as previously described [56] except that the fixation duration was increased to 35 mins. Primary antibodies used include chicken anti-GFP (Catalog #GFP-1020, Aves Labs, Tigard, Oregon; 1:500–1000), rabbit anti-DsRed (Catalog #632392, Takara Bio USA, Inc., Mountain View, CA; 1:250), rabbit anti-Dpn (1:500) (a gift from Dr. Y. N. Jan) [57], guinea pig anti-Ase (a gift from Dr. Y.N. Jan, 1:5000) [54], rabbit anti-PntP1 (a gift of Dr. J. B. Skeath; 1:500) [58], and rabbit anti-Tll (a gift from of Dr. J. Reinitz). Secondary antibodies conjugated to Daylight 405 (1:300–400), Daylight 488 (1:100), Cy3 (1:500), Rhodamine Red-X (1:500), Daylight 647 (1:500) or Cy5 (1:500) used for immunostaining are from Jackson ImmunoResearch (West Grove, PA). The F-actin probe of phalloidin labeled with Alexa Fluor 555 (Catalog #A34055, Thermo Fisher Scientific, Waltham, MA) was used to label type I or type II NB lineages. Images were collected using a Carl Zeiss LSM780 confocal microscopy and processed with Adobe Photoshop. Student's T-test or Mann Whitney U test (when data were not normally distributed) was used for statistical analyses between two groups of samples. ANOVA followed by multiple comparisons using post hoc tests were used for statistical analyses for the comparison of multiple groups of samples. All the numerical data used to generate graphs and corresponding statistical analyses were summarized in S1 Data. Graphs were generated with Microsoft Excel or GraphPad Prism 8.

## Plasmid construction and transgenic fly generation

To generate the chimeric repressors with or without the Ets domain, the sequence encoding SV40-NLS was synthesized (Integrated DNA Technologies, Inc., Coralville, Iowa) and cassettes of the repressor domain of Engrailed (aa. 1–298) (EnR) or the Ets domain of PntP1 were amplified by CloneAmp HiFi PCR Premix (Catalog# 639298, Takara Bio., Mountain View, CA) from genomic DNAs of the *UAS-PntP1* line. To generate chimeric activators containing the Ets domain, the VP16 activation domain was amplified from genomic DNAs of a VP16AD transgenic line, and sub-fragments of PntP1 shown in S3A Fig were amplified from the *UAS-PntP1* line. These cassettes were cloned into the pUAST vector in the order as designed in S2A or S3A Figs using corresponding restriction sites. Primers or oligos used are listed below.

*SV40NLS_sense: AATTCATGCCAAAAAAGAAGAGAAAGGTAA*
*SV40NLS_Antisense: GATCTTACCTTTCTCTTCTTTTTTGGCATG*
*PntP1Ets_F: AAATCTCGAGTTCACGGGATCGGGTCCCATT*
*PntP1Ets_R: ACAATTCTAGACTAATCACAGACAAAGCGGTAGACATATCG*
*EnR_F: AAATAGATCTATGGCCCTGGAGGATCGCT*
*EnR_R: AAATCTCGAGGGATCCCAGAGCAGATTTCTCTGGA*
*VP16AD_F: AAATAGATCTGCCCCCCCCGACCGATGTCAGCCT*
*VP16AD_R: AAATCTCGAGCCCACCGTACTCGTCAATTCCAAGGGCAT*
*PntP1(1/2N) _F1: AAATAGATCTACTAGTATGCCGCCCTCTGCGTTTTTAGTG*
*PntP1(1/2N) _R1: AAATAGATCTCGGCTGCTGCGATTGCTGCT*
*PntP1(1/2N) _F2: AAATGCTAGCATGCCGCCCTCTGCGTTTTTAGTG*
*PntP1(1/2N) _R2: AAATGCTAGCACTAGTCGGCTGCTGCGATTGCTGCT*
*PntP1(1/2C) _F: AAATGCTAGCACTGTCAATGGCAGCGGTAG*
*PntP1(1/2C_Ets) _R (same as PntP1Ets_R): ACAATTCTAGACTAATCACAGACAAAGCG
GTAGACATATCG*

The derived constructs were verified by sequencing (Genewiz, South Plainfield, NJ) and injected into *Drosophila* embryos by the Rainbow Transgenic Flies Inc (Camarillo, CA) or BestGene Inc (Chino Hills, CA). The transformants were generated following standard P-element transposon protocols.

The gateway technology was used to generate *pDes-(ase)enhancer-GAL4* constructs as shown in Fig 1A. Briefly, individual *ase* enhancer fragments were amplified by PCR from genomic DNAs and incorporated into pDONOR221 vector by BP reactions to get entry clones. Then LR reactions were performed to transit the *ase* enhancer fragments from the entry clones into the destination vector *pBPGUW* to get desired *pDes-(ase)enhancer-GAL4* constructs. Primers used to amplify the attB sequence flanking the *ase* enhancer fragments are listed below (the underlined are parts of attB1 in forward (F) oligos and attB2 in reverse (R) oligos).

*(-1,734 ~ -1,060)_F:* <u>GGGGACAAGTTTGTACAAAAAAGCAGGCT</u>GGATCCAGTATGTT
TCCACG

*(-1,295 ~ -1,060)_F:* <u>GGGGACAAGTTTGTACAAAAAAGCAGGCT</u>ATGCGTCGTCAAAG
TGGGAC

*(-1,287 ~ -1,060)_F:* <u>GGGGACAAGTTTGTACAAAAAAGCAGGC</u>TTCAAAGTGGGACGC
AACCGA

*(-1,282 ~ -1,060)_F:* <u>GGGGACAAGTTTGTACAAAAAAGCAGGCT</u>GTGGGACGCAACCG
AGTCAA

*(-1,279 ~ -1,060)_F:* <u>GGGGACAAGTTTGTACAAAAAAGCAGGCT</u>GGACGCAACCGAGT
CAAATC

*(-1,276 ~ -1,060)_F:* <u>GGGGACAAGTTTGTACAAAAAAGCAGGCT</u>CGCAACCGAGTCAA
ATCCTC

*(-1,212 ~ -1,060)_F:* <u>GGGGACAAGTTTGTACAAAAAAGCAGGCT</u>ACTCCTCAGTACGC
ACAAGC

*(-1,180 ~ -1,060)_F:* <u>GGGGACAAGTTTGTACAAAAAAGCAGGCT</u>TTCTTTGAGAGCTC
GTCTGCATA

*(-1,138 ~ -1,060)_F:* <u>GGGGACAAGTTTGTACAAAAAAGCAGGCT</u>CATCCTGGATCAAA
AACCGGTA

*(-1,087 ~ -1,060)_F:* <u>GGGGACAAGTTTGTACAAAAAAGCAGGCT</u> CGGTTATCCTGCGC
TCAAGT

*(-1,734 ~ -1,060)_R:* <u>GGGGACCACTTTGTACAAGAAAGCTGGGT</u>GGAAAAGGACTTGA
GCGCAG

*(-1,212 ~ -1,166)_R: GGGGACCACTTTGTACAAGAAAGCTGGGTCGAGCTCTCAAAGA* *AAAAGGAGT*

*(-1,212 ~ -1,138)_R: GGGGACCACTTTGTACAAGAAAGCTGGGTGCCACGTCCTCATT* *CCTCATTAT*

*(-1212 ~ -1,086)_R: GGGGACCACTTTGTACAAGAAAGCTGGGTCGGCGGGGAGAAG* *AAACTTG*

For generating the *pDes-(ase)enhancer-GAL4* construct containing mutations in Tll_site_R, the primer set of *Tll_site_R_Mutant_F* (see the sequence below) and *(-1,734 ~ -1,060)_R* were used to amplify the *ase* fragment containing mutations in Tll_site_R and then the primer set of *(-1,295 ~ -1,060)_F* and *(-1,734 ~ -1,060)_R* were used to add attB sites.

Tll_site_R_Mutant_F: ATGCGTCGTCAAAGTGGGACGCAACCtAtctAcATCCTCTAG GACAACAAAGGACGCCGA

Lower cases indicate mutations in Tll_site_R.

All the *pDes-(ase)enhancer-GAL4* constructs were integrated into the same attp2 docking sites on the 3rd chromosome (line #8622, BDSC) through the φ31 integration system by Rainbow Transgenic Flies, Inc (Camarillo, CA) or BestGene Inc (Chino Hills, CA).

## Electrophoretic mobility shift assays (EMSAs)

EMSA experiments were performed as previously described [15]. Briefly, Tll or PntP1 coding region was amplified using the CloneAmp HiFi PCR Premix (Catalog# 639298, Takara Bio., Mountain View, CA) from a cDNA library and cloned into pcDNA3.1/His expression vectors (Catalog #V38520, Life Technologies Co., Grand Island, NY). Proteins were expressed from the pcDNA3.1/His construct by the TNT T7 Quick Coupled Transcription/Translation kit (Catalog #L1170, Promega Co., Madison, WI) according to manufacturer's instructions. Empty pcDNA3.1/His vectors were used as negative controls. Previously confirmed 25-nt Tll bound sequence from the *kr* enhancer [25] and a 27-nt PntP1 binding sequence [16] from the *erm* enhancer fragment *R9D11* were chosen as positive probes, which were labeled with Cy5 at the 5'-end of both strands. Other Cy5 labelled probes with sequences from either the *ase* enhancer or the *tll* enhancer were shown in S1A and S6A Figs, respectively. The binding reactions were performed by incubating 0.05 pmol of Cy5-probes with 3μl TNT-T7 expressing product and/or cold competitors in an amount of 50, 200 or 500-fold over Cy5-probes as indicated. All probes and competitors were synthesized by Integrated DNA Technologies (Coralville, IA) and annealed to form dsDNAs. To verify the binding specificity of the probes, the competitors containing the same sequence as the Cy5-probes were used as specific competitors or anti-Xpress antibody (Catalog #R910-25, Thermo Fisher Scientific, Waltham, MA) was added to the binding reactions for detecting a super-shifted band. To test the binding between two putative Tll binding sites (Tll_site_L and Tll_site_R) in the *ase* enhancer region and Tll protein, a series of competitors (competitors 2–9) with sequences containing the wild type sites and/or mutated sites of Tll_site_L or Tll_site_R along with the flanking nucleotides were generated and added in the amounts as indicated in the binding reactions. To test the binding between the putative PntP1 binding sites in *tll* enhancer *R31F04* and PntP1 proteins, a series of competitors containing the wild type (competitors T1-T7) or mutated (competitors T1Mu-T7Mu) PntP1 binding sites along with the flanking sequences were generated and added in the amounts as indicated in the binding reactions. Two other sites in the *tll* enhancer (competitors neg#1 and neg#2) and competitor 1 were also used as negative controls. Xpress-labeled proteins were preincubated with competitors for 15 mins at room temperature followed by 25-min incubation with Cy5-probes. 10μl of EMSA binding reactions were loaded on 5% non-denaturing polyacrylamide Mini-PROTEAN TBE gels (Catalog #4565015, Bio-Rad

Laboratories, Hercules, CA) for electrophoretic analyses. The electrophoresis was run for 25 mins or 1 hr at a constant current of 30mA. The gels were scanned and analyzed with Chemi-Doc XRS (Bio-Rad Laboratories, Hercules, CA). Sequences of the oligonucleotides in all competitors are summarized in Figs 3A, 5C, and S1A.

## Chromatin immunoprecipitation (ChIP) and quantitative PCR (qPCR)

ChIP experiments were performed as described before [43]. Briefly, *Drosophila* larvae with the genotype of *UAS-mCD8-GFP, UAS-Dcr2/UAS-HA-PntP1; pntP1-GAL4/UAS-Brat RNAi* were raised at 30˚C till late 3rd instar stages for brain dissection. Around 100 brains, which contained a total of approximately $2 \times 10^6$ type II NBs, were used for every single IP reaction. Brains were collected and fixed in 1.8% formaldehyde solution for 20 mins at room temperature. The fixation was ended by adding Glycine with a final concentration of 250mM at room temperature for 5 mins. Fixed samples were washed with cold 1X DPBS (Catalog #14190144, Thermo Fisher Scientific) and homogenized in SDS lysis buffer (1% SDS, 10mM EDTA, 50mM Tris-HCl pH8) with proteinase inhibitors (Catalog #78430, Thermo Fisher Scientific) and 1mM AEBSF (Catalog #78431, Thermo Fisher Scientific). In order to fragmentize DNAs to a range of 200-700bps, around 30 brains in 110µl lysis buffer in one microtube-130 AFA Fiber Pre-Slit Snap-Cap (Catalog #520045, Covaris, Inc. Woburn, MA) were sonicated on a Covaris ME220 sonicator (Catalog # 500295, Covaris, Inc.) using following settings: 50W Peak incident power, 10% Duty factor, 200 Cycles per burst (Cpb) for 90S and repeated for 3 times with 20S interval. Sonicated lysate was pooled and centrifuged to remove debris, and diluted with ChIP dilution buffer (1.1% Triton X-100, 1.2 mM EDTA, 16.7 mM Tris–HCl pH8, 167 mM NaCl) containing proteinase inhibitors and 1mM AEBSF to 1 brain per 10µl. 1ml diluted lysate were used for each IP and 200ul (20%) of diluted lysate was saved for Input. Samples were pre-cleared with Dynabeads Protein A and Dynabeads Protein G (Catalog #10001D and #Catalog #10003D, Thermo Fisher Scientific) for 1hr at 4˚C first and then incubated with rabbit anti-PntP1 antibody (a gift of J. B. Skeath; 1:200) [58] or 3µg/µl isotypic rabbit IgG (Catalog #10500C, Thermo Fisher Scientific) overnight at 4˚C. Overnight pre-blocked Dynabeads Protein A and G by 2mg/ml BSA, 0.5mg/ml salmon tests DNAs (Catalog #D7656, Sigma-Aldrich Co, St. Louis, MO) in ChIP dilution buffer were added for extra 4 hrs of incubation at 4˚C. Beads were then washed three times with low salt wash buffer (0.1% SDS, 1% Triton X-100, 2mM EDTA, 20mM Tris–HCl pH8, 150 mM NaCl), twice with high salt wash buffer (0.1% SDS, 1% Triton X-100, 2mM EDTA, 20mM Tris–HCl pH8, 500 mM NaCl), three times with LiCl wash buffer (1% NP40, 1% sodium deoxycholate, 1mM EDTA, 10mM Tris–HCl pH8, 0.25M LiCl), twice with TE buffer (10mM Tris-HCl pH8, 1mM EDTA), and eluted with freshly made elution buffer (1% SDS, 0.1M NaHCO$_3$). Samples were incubated at 65˚C overnight for reversing cross-linking of the chromatin–protein complex and degrading RNAs with 100µg/ml RNase A (Catalog # 12091021, Thermo Fisher Scientific), and then treated with 100µg/ml proteinase K at 45˚C for 1 hr. DNA samples were purified with GeneJET PCR Purification Kit (Catalog #K0701, Thermo Fisher Scientific) following manufacture's manuals before being used for qPCR. The fold enrichment of anti-PntP1 antibody was normalized to IgG controls and calculated as ΔΔCt of three biological replicates. The numerical data and corresponding statistical analyses were summarized in S1 Data.

Primer sets targeting PntP1 binding loci in *R31F04* fragment were:

Pnt_13F: CCTACCATGGGCATGAGGTT

Pnt_13R: ACTCTTACCGAATTCGCCCC

Pnt_1F: TGGAAATGGAAGCAGCGACT

Pnt_1R: CCTCCAAGATTTGGGCCACT

Pnt_3F: CCCGCCGTGTAATTATTGCG
Pnt_3R: CCGAGAACGAGATCCACAGG
Pnt_4F: CAAGCCTCTTGGGGTTCCAT
Pnt_4R: AGTGCAAACAGACAGGGGAG
Pnt_12F: CTCCCCTGTCTGTTTGCACT
Pnt_12R: TGGCGCCAAGAAGACAAAGA
Primer sets targeting negative loci in *R31D09* fragment were:
Neg#3F : CAGAGAGCTGTTCCCACGTT
Neg#3R: TTCTGTCTTCGGGATCAGCG
Neg#5F: GGTATTCAACCCCTGCTGCT
Neg#5R: GCTGCCATTATTGCCGCTTT
Primer sets targeting negative loci in *ase enhancer* were:
AseF: CTTGCAGTGCACGAAAGGC
AseR: CAACGCTTGTGCGTACTGAG

## Supporting information

**S1 Fig. Tll proteins directly bind to the *cis*-repressive elements in the *ase* regulatory region.**
Open arrowheads point to free Cy5-probes at the bottom of gels, while filled arrowheads point
to the lagged bands of the complexes of Tll-probes or super-shifted bands of the complexes of
Xpress Ab-Tll-probes. Numbers below the lane numbers indicate the ratio of the competitor
to the probe. The indicated competitors are the same as those shown in Fig 3. The lanes are
numbered following the lane numbers in Fig 3. (A) The sequences of probes Cy5-probe2,
Cy5-probe3, and Cy5-probe4 containing Tll_site_L, Tll_site_R, or both are shown. (B) No
obvious binding is detected between Xpress-Tll and Cy5-Probe2 (lane #23), which contains
Tll_site_L, possibly because the band is below the detection threshold. (C) The binding
between Xpress-Tll and Cy5-probe3 and its competition by indicated competitors. Probe3
contains the sequence of Tll_site_R. Specific binding between Tll and Cy5-probe3 is detected
in lane #25 and verified in lane #34 by the presence of a super-shifted band when the Xpress
antibody is present. Both competitor 4, which has the same sequence as Cy5-probe3, and com-
petitor 2, which contains the sequence of Tll_site_L, can compete with Cy5-probe3 for the
binding with Xpress-Tll in a dose-dependent manner (lanes #26, #28, #30, and #32), but com-
petitors (competitors 5 and 3) that contains only a mutated Tll_site_R or a mutated Tll_site_L
cannot (lanes #27, #29, #31, and #33). (D) The binding between Xpress-Tll and Cy5-probe4
and its competition by indicated competitors. Probe4 contains both Tll_site_L and Tll_site_R.
Cy5-probe4 binds to Xpress-Tll (lane #37). Both competitor 6 (lane #38), which contains the
same sequence as Cy5-probe4, and competitor 1 (lane #42), which contains a Tll binding site
from the *kr* enhancer, strongly competes with Cy5-probe4 for the binding with Xpress-Tll.
Competitor 7 that contains a mutated Tll_site_L and the wild type Tll_site_R (lane #39) also
strongly competes with Cy5-probe4 for binding with Xpress-Tll proteins. Competitor 8 that
contains the wild type Tll_site_L and a mutated Tll_site_R can only weakly compete with the
Cy5-probe4 for the binding (lane #40). Competitor 9, in which both Tll_site_L and Tll_site_R
are mutated, can hardly compete with probe4 for the binding (lane #41). (E) Electrophoretic
separation of two distinct bands of the Xpress-Tll-Cy5-probe3 (lane #43) or Xpress-Tll-Cy5-p-
robe4 (lane #44) complexes after the gel has been run for 1 hr at a constant current of 30mA.
(TIF)

**S2 Fig. The EnR-Ets chimeric repressor antagonizes endogenous PntP1's function.** NB line-
ages are labeled with mCD8-GFP (in green) driven by *insc-GAL4* for type I NB lineages (B-F)
or by *pntP1-GAL4* for type II NB lineages (G-L), and counterstained with anti-Dpn (in red) or

anti-Ase (in blue) antibodies. White arrows point to some Ase$^+$ Dpn$^+$ type I NBs or Ase$^-$ Dpn$^+$ type II NBs as examples. White arrowheads point to Ase$^+$ Dpn$^+$ mINPs or mINP-like cells. Yellow arrowheads point to Ase$^+$ Dpn$^-$ GMCs. Dashed lines highlight representative NB lineages enlarged in insets. In this and all the following figures, only images from one brain lobe or the thoracic segments of VNCs are shown. The brain lobes are oriented so that the midline is to the right and the anterior side of the brain up. Scale bars equal 50μm. (A) Schematic diagrams of constructs used to express PntP1, NLS-EnR-Ets, NLS-Ets, or NLS-EnR. (B-B') VNCs contain only Ase$^+$ Dpn$^+$ type I NBs, which directly produce Ase$^+$ Dpn$^-$ GMCs. (C-C') Misexpressing *UAS-pntP1* in type I NBs represses Ase expression and induces Ase$^+$ Dpn$^+$ mINP-like cells. (D-F') Misexpressing *UAS-NLS-EnR-Ets* (D-D'), *UAS-NLS-Ets* (E-E') or *UAS-NLS-EnR* (F-F') in type I NBs does not repress Ase expression or induce any mINP-like cells. (G-H') A wild type (G-G') or *UAS-pntP1* overexpressing brain lobe (H-H') has eight type II NB lineages, each of which contains an Ase$^-$ Dpn$^+$ type II NB and multiple Ase$^+$ Dpn$^+$ mINPs. (I-I') Type II NBs with *UAS-NLS-EnR-Ets* misexpression ectopically expresses Ase and directly produces Ase$^+$ Dpn$^-$ GMCs instead of INPs. Note that *UAS-NLS-EnR-Ets* misexpression also leads to the generation of supernumerary type II NB (see white arrows) lineages. (J-K') Misexpressing *UAS-NLS-Ets* (J-J') or *UAS-NLS-EnR* (K-K') does not affect the normal development of type II NB lineages, including the repression of Ase, the number of type II NBs, or the composition of cell types. (L-L') Knocking down PntP1 by expressing *UAS-pnt RNAi* results in ectopic Ase expression in type II NBs, loss of INPs and generation of supernumerary type II NBs. (M-N) Quantifications of the percentage of Ase$^-$ type I NBs and the percentage of type I NB lineages with mINPs in VNCs (M), and the percentage of Ase$^+$ type II NBs, the percentage of type II NB lineages with mINPs, and the number of total GFP$^+$ type II NBs per brain lobe (N) with indicated genotypes. $^{***}$, $P < 0.001$. Values of the bars are mean ± SD. The numbers on top of, or within each bar in graphs in this and all following figures indicate the number of samples examined.

(TIF)

**S3 Fig. PntP1 functions as a transcriptional activator to repress Ase expression and promote INP generation.** In all images, type I NB lineages in VNCs are labeled with mCD8-GFP (in green) driven by *insc-GAL4*, and counterstained with anti-Dpn (in red) or anti-Ase (in blue) antibodies. White arrows point to representative type I NBs in which Ase is expressed in (B-B', D-D', E-F', and H-H') or those in which Ase is repressed in (C-C' and G-G'). White arrowheads point to induced Ase$^+$ Dpn$^+$ mINP-like cells and yellow arrowheads point to Ase$^+$ Dpn$^-$ GMCs. Dashed lines highlight representative NB lineages enlarged in insets. Scale bars equal 50μm. (A) Schematic diagrams of constructs used to express PntP1, or sub-fragments of PntP1 with or without the fused VP16AD domain. (B-B') VNCs contain only Ase$^+$ Dpn$^+$ type I NBs, which directly produce Ase$^+$ Dpn$^-$ GMCs. (C-C') Misexpressing *UAS-pntP1* represses Ase expression in type I NBs and induces Ase$^+$ Dpn$^+$ mINP-like cells. (D-F') Misexpressing chimeric activators NLS-VP16AD-Ets (D-D'), NLS-pntP1(1/2N)-VP16AD-Ets (E-E') or NLS-VP16AD-pntP1(1/2N)-Ets (F-F') does not suppress Ase expression in type I NBs. (G-H') Misexpressing the chimeric activator NLS-VP16AD-pntP1(1/2C)-Ets represses Ase expression and induces Ase$^+$ Dpn$^+$ mINP-like cells (G-G'), while misexpressing NLS-pntP1(1/2C)-Ets does not (H-H'). (I) Quantifications of the percentage of type I NBs with Ase being suppressed or with mINP-like cells in VNCs expressing indicated chimeric proteins or wild type PntP1. Values of the bars are mean ± SD. $^{***}$, $P < 0.001$. (J) Summary of phenotypes resulting from the expression of PntP1, or indicated chimeric proteins in type I NB or type II NB lineages. (K) A diagram of functional domains of the PntP1 protein based on phenotypic analyses of the chimeric activator proteins. The potential activation domain is likely localized in the aa.1-255

region of PntP1, while the region of aa.256-511 is probably necessary for PntP1's activity, potentially involved in recruiting co-factors, and the Ets domain (aa.512-597) is required for binding to target DNAs.
(TIF)

**S4 Fig. Knockdown of PntP1 and Tll overexpression synergistically promote the generation of supernumerary type II NBs.** In all images, type II NB lineages are labeled with mCD8-GFP (in green) driven by *pntP1-GAL4* and counterstained for Ase (in red) and Dpn (in blue). Arrows point to type II NBs, white arrowheads point to Ase$^+$ Dpn$^+$ mINPs, and yellow arrowheads point to Ase$^+$ Dpn$^-$ GMCs. Scale bars equal 50μm. (A) A wild type brain lobe contains eight type II NBs, which are associated with their mINPs and GMCs. (B) Knockdown of PntP1 results in ectopic Ase expression in type II NBs (purple arrows filled with white), which directly produces GMCs at the expense of mINPs, and a slight increase in the total number of GFP labeled type II NBs. mINPs are also reduced or depleted in the lineages without the ectopic Ase expression in the NBs (e.g., white arrows). (C) Overexpressing Tll alone promotes the generation of supernumerary type II NBs (white arrows) but a large number of mINPs are still generated. (D) Simultaneous knockdown of PntP1 and overexpression of Tll leads to a dramatic increase in the number of type II NBs and near complete depletion of mINPs. (E-F) Quantifications of the number of type II NBs per lobe (E) or the total number of mINPs per lobe (F) in the brains with indicated genotypes. Values of the bars are mean ± SD. The numbers on each bar represent the number of brain lobes examined. ***, $P < 0.001$.
(TIF)

**S5 Fig. Misexpressing PntP1 activates Tll expression in type I NBs.** Type I NB lineages in the VNC are labeled with mCD8-GFP (in green) driven by *insc-GAL4*. The VNCs are counterstained with anti-Tll (in red), anti-Ase (in blue) antibodies. White arrows point to some representative type I NBs. Scale bars equal 10μm. (A-A") Tll expression is not detected in type I NBs in VNCs. (B-B") Misexpressing *UAS-PntP1* activates ectopic Tll expression in all type I NBs and represses Ase expression.
(TIF)

**S6 Fig. PntP1 directly binds to the putative binding sites in *tll* regulatory regions.** (A) Top: the sequence of Cy5-probe6 that contains the putative PntP1 binding site #4 shown in green. The green arrow indicates the orientation of the consensus PntP1-binding 5'-GGAA/T-3' core sequence. The underlined sequence in the probe indicates the GGAA core sequence of the PntP1 binding motif. Bottom: binding of Cy5-probe6 with Xpress-PntP1 and its competition by the indicated competitors. The sequences of the competitors are the same as shown in Fig 5. Lanes are numbered following the lane numbers in Fig 5. Xpress-PntP1 proteins bind to Cy5-probe6 (lane #28). The super-shifted band in lane #25 indicates a further retardation on the mobility of the complex of Xpress Ab-PntP1-Cy5-probe6, when the anti-Xpress Ab is present. Both competitor T4 (lane #29), which contains the same sequence as Cy5-probe6, and competitor P1 (lane #45) can competes with Cy5-probe6 for the binding with Xpress-PntP1, but competitor T4Mu, which contains mutated PntP1 binding site #4 (lane #30) cannot. Competitors T1, T2, T3, T5, T6, or T7, which contains PntP1 binding sites #1, 2, 3, 5, 6, or 7 (corresponding to lanes #31, #33, #35, #37, #39, or #41), respectively, can competes with Cy5-probe6 for the binding to PntP1 to various extent, while competitors T1Mu, T2Mu, T3Mu, T5Mu, T6Mu, or T7Mu, which contains mutated binding site #1, 2, 3, 5, 6, or 7 (corresponding to lanes #32, #34, #36, #38, #40, or #42) or non-specific competitors neg#1 (lane #43) or neg#2 (lane #44), cannot. Open arrowheads point to free probes while filled arrowheads point to bands of the PntP1-probe complexes or the super-shifted bands of the Xpress Ab-PntP1-probe

complexes. Numbers below the lane numbers indicate the ratio of the competitor to the probe.
(TIF)

**S7 Fig. PntP1 unlikely activates additional downstream targets in parallel of Tll to suppress Ase.** Type II NB lineages are labeled with mCD8-GFP (in green) driven by *pntP1-GAL4*
or *R31F04-GAL4* and Type I NBs are labeled with mCD8-GFP (in green) driven by *insc-GAL4*.
The brains and VNCs are counterstained with anti-Ase, anti-Dpn, and/or anti-PntP1 antibodies. White arrows point to type II NBs in (A-E") or type I NBs with ectopic PntP1 expression
in (F-$G_1$"). Open arrows in (F-$G_1$") point to type I NBs without the ectopic PntP1 expression.
Scale bars equal 10μm in A-G or 50μm in H-I. (A-C") Expression of *UAS-PntP1* driven by
*pntP1-GAL4* increases PntP1 levels in wild type type II NBs lineages (A-A'), and restores
PntP1 expression in 23.2% of Tll knockdown type II NB lineages to levels comparable to those
in wild type type II NB lineages (C-C") but not in other 76.8% of Tll knockdown type II NB
lineages (B-B"). However, Ase remains ectopically expressed in Tll knockdown type II NBs
that have restored PntP1 levels (C-C"). (D-E") Expression of *UAS-tll-shRNA* driven by
*R31F04-GAL4* leads to the loss of PntP1 and ectopic Ase expression in type II NBs (D-D").
Expression of *UAS-PntP1* driven by *R31F04-GAL4* restores PntP1 expression in 95.8% of Tll
knockdown type II NBs. However, Ase remains ectopically expressed in all Tll knockdown
type II NBs (E-E"). (F-$G_1$") PntP1 is never expressed in wild type type I NBs in the VNC (F-F")
and expressing *UAS-Tll* induces PntP1 expression in a subset of type I NBs but a majority of
Tll misexpressing type I NBs still do not express PntP1 (G-G"). ($G_1$-$G_1$") show an enlarged
view of the area highlighted with a dotted square in (G-G"), respectively. (H-I") Expressing
*UAS-Tll* suppresses Ase expression and promotes the generation of supernumerary type I NBs
(H-H"), and simultaneous knockdown of PntP1 does not restore Ase expression or inhibit the
generation of supernumerary type I NBs (I-I"). (J-K) Quantifications of PntP1 staining intensity in type II NBs (J) and the percentage of Ase$^+$ type II NBs (K) in animals with indicated
phenotypes. The PntP1 intensity in wild type NBs is normalized as 1 in (J). Values represent
mean ± SD. Numbers on each bar indicate the number of type II NBs (J) or brain lobes (K)
examined. ***, $P < 0.001$. NS, not significant. (L-M) Quantifications of the percentage of
PntP1$^+$ (L) and Ase$^+$ (M) type II NBs in animals with indicated phenotypes. Values of the bars
are mean ± SEM. Numbers on each bar indicate the number of brains examined. ***,
$P < 0.001$. NS, not significant. (N) Quantifications of the percentage of Ase$^+$ type I NBs (N) in
animals with indicated phenotypes. Values of the bars are mean ± SD. Numbers on each bar
indicate the number of VNCs examined. ***, $P < 0.001$. NS, not significant.
(TIF)

**S1 Data. All the numeric data used to generate the graphs and tests for statistical analyses.**
(XLSX)

## Acknowledgments

We thank Drs. Y. N. Jan, J. B. Skeath, A. H. Brand, C.Y. Lee, J. Reinitz, the Bloomington *Drosophila* Stock Center, Vienna *Drosophila* Resource Center, Zurich ORFeome Project, and *Drosophila* Genomics Resource Center for antibodies, plasmids and fly stocks; Drs. F. Pignoni and
A. S. Viczian for sharing research facility; Dr Y. Zhang for technique support; Drs. R. T. Matthews, F. Pignoni, M. E. Zuber, J. Amack, K. E. Lewis, X Li, M. Connell, S. J. Neal for thoughtful discussion and comments.

## Author Contributions

**Conceptualization:** Rui Chen, Sijun Zhu.

**Data curation:** Rui Chen.

**Formal analysis:** Rui Chen.

**Funding acquisition:** Sijun Zhu.

**Investigation:** Rui Chen.

**Methodology:** Rui Chen, Xiaobing Deng.

**Project administration:** Sijun Zhu.

**Resources:** Rui Chen, Xiaobing Deng, Sijun Zhu.

**Software:** Rui Chen.

**Supervision:** Sijun Zhu.

**Validation:** Rui Chen.

**Visualization:** Rui Chen.

**Writing – original draft:** Rui Chen.

**Writing – review & editing:** Rui Chen, Sijun Zhu.

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
