## [Decision Letter · Decision Letter 0]

15 Dec 2021

Dear Dr Zhu,

Thank you very much for submitting your Research Article entitled 'The Ets protein Pointed P1 represses Asense expression in type II neuroblasts by activating Tailless' to PLOS Genetics.

The manuscript was fully evaluated at the editorial level and by independent peer reviewers. The reviewers appreciated the attention to an important topic but identified some concerns that we ask you address in a revised manuscript.

We therefore ask you to modify the manuscript according to the review recommendations. Your revisions should address the specific points made by each reviewer.

[LINK]

Yours sincerely,

Hongyan Wang, Ph.D.

Associate Editor

PLOS Genetics

Gregory P. Copenhaver

Editor-in-Chief

PLOS Genetics

Reviewer's Responses to Questions

**Comments to the Authors:**

Reviewer #1: Asymmetric cell division is a mechanism to generate cell diversity, while maintaining a progenitor cell. Neuronal stem cells utilize ACD to form neuronal subtypes either directly or through the formation of intermediate progenitor cells. The mechanisms determining which pathway is used are incompletely understood. Here, Chen et al., use Drosophila neural stem cells, neuroblasts hereafter, to shed light on a transcriptional cascade that is used to specify type I, or intermediate progenitor forming (type II) neuroblasts. The authors provide evidence that the Ets protein Pointed induces the transcription of the transcriptional repressor Tailless. Tailless represses Asense expression in type II neuroblasts, thereby creating a subset of neuroblasts that are molecularly and functionally distinct from type I neuroblasts.

This is an interesting and detailed study that will be of interest to the neural stem cell field. The conclusions are predominantly supported by the presented data but I have two major concerns that need to be addressed.

(1) In my opinion, the use of chimeric constructs in Figure 1 and 2 to define whether Pointed acts as a transcriptional repressor vs. activator is borderline convincing mostly because the constructs are highly artificial and are inconclusive if they provide negative results. Did the authors consider generating Pointed deletion constructs, expressed in a pointed mutant background to define repressor vs. activator domains of pointed? The ase enhancer deletion analysis on the other hand is straightforward and convincing. I suggest moving the data shown in figure 1 and 2 to the supplementary figures and start the story with figure 3.

(2) Similarly, I find the logic of the experiments in figure 4 hard to follow and the data inconclusive. Since these constructs are expressed in a wild type background, how can the authors exclude the possibility that endogenous Tailless modulates the expression of the reporters? It would be more convincing to monitor ase expression in tailless mutants or under conditions of tailless misexpression in combination with the generated deletion constructs shown in figure 3 to define the tailless responsive element genetically.

Minor critiques:

(3) Figure 2E, H, F, D lack high magnification inserts. These are necessary to verify the claims.

Reviewer #2: The paper “PntP1 represses Asense by activating Tailless” aims to investigate the molecular mechanism underlying Ase suppression by PntP1 in type II NBs during the larval development. By constructing artificial proteins containing the Ets domain of PntP1 and a repressor/activator domain of another proteins, the authors suggested that instead of a direct transcriptional suppressor, PntP1 acts as an activator to indirectly repress ase expression. To identify the downstream effector of PntP1, a bottom-up approach was exploited in which the authors first explored the enhancer region of ase. By combining bioinformatics and molecular methods, the author mapped multiple sites in ase enhancer sequence required for the expression of Ase in type II or in all NBs. In a mapped sequence with repressive elements, a Tll binding motifs were identified in addition to another neighboring downstream of the repressive elements. Although EMSA results showed that both motifs in ase enhancer regions could bind to Tll in vitro, they appeared to have different affinity. Furthermore, PntP1 was demonstrated to be both necessary and sufficient for the expression of Tll. In addition, EMSA and ChIP-qPCR were utilized to show that PntP1 can directly bind to the enhancer region of tll in vitro and in vivo. On the other hand, Tll necessitates the maintenance of PntP1 expression. Interestingly, when PntP1 is expressed and Tll is knocked down in type I NBs, ectopic Ase expression remains. As such, it was suggested that PntP1 did not activate any other downstream effector in parallel to Tll to suppress Ase expression.

Overall:

The methods used were appropriate to address the main research question although some details need further explanations. The manuscript provides excellent abstract and introduction that is clear and captured the essential knowledge necessary to understand further work. The provided data mostly supports the conclusions. However, some aspects of data visualization are a bit confusing. Additionally, statistical analysis this manuscript may require extra consideration.

Minor:

- Methodology

o EMSA: the first part in which the authors mentioned about this technique was a bit unclear. Therefore, it may need 1-2 sentence to explain how to read the data provided, for example, what to expect if the hypothesis is correct.

o In the last section in which the authors investigated if PntP1 also regulates other downstream effectors to suppress Ase expression in parallel to Tll, pnt-Gal4>UAS-PntP1 in addition to Tll KD was used. In the results, UAS-PntP1 failed to restore PntP1 expression in this condition, likely due to the downregulation of pnt-Gal4 expression upon Tll KD due to the Tll-PntP1 positive feedback loop. Based on this and later in type I NBs with restored PntP1 expression, the authors concluded that Tll is the only target of PntP1 that controls Ase expression. Is there any type II NB-specific Gal4 that can drive UAS-PntP1 without being affected by Tll expression? If so, using this Gal4 alternatively of pnt-Gal4 maybe able to strengthen the argument.

o The authors only showed that Tll binds to ase enhancers in vitro via EMSA. However, in vivo evidence is lacking here. As Hakes and Brand (2020) previously performed TaDa to show that Tll binds to ase in vivo, the authors can either use another method (ChIP-qPCR) to validate or refer to this previous literature.

o ChIP-qPCR: Brat KD was used to enrich the type II NBs for ChIP-qPCR. Nevertheless, it was not discussed such as how using Brat KD may affect the results (e.g., effect on PntP1, Tll and their bindings)

- Results and discussion

o The paper of Hakes and Brand (2020) was mentioned in the section regarding Tll-PntP1 feedback loop, referring to when Tll is misexpressed, there is ectopic expression of PntP1. I did not fine this information in the original publication. Thus, the reference needs to be double-checked

o PntP1 was claimed to activate only Tll as a downstream effector to inhibit Ase expression. However, I think PntP1 can regulate many factors among which Tll is the key regulators. Others may still exit and regulate Ase expression, cooperatively with Tll. As such, the conclusion here may be a bit too far.

o Limitations of the methods were not much discussed

- Data representations

o In figure 4: figure 4H was mentioned at the beginning of the section ‘Tll suppresses Ase expression directly by binding to ase enhancer). However, in the figure, it was placed after the others. Although it is understandable that the authors want to show the figures as a concluding model of the section, it causes confusion, the figures should be cited in chronological order to enhance readability.

o Figure captions did not mention what data represents (e.g., mean ± SD)

o Figure 8H which showed the PntP1 intensity in type II NBs, if the data is normalized to the WT, it should be indicated in the axis label like other graphs in the manuscript (This is the only one that employed dot plot instead of bar graph).

- Statistical analysis

o Student’s t-test was used throughout the manuscript regardless the sample number. As there are more than two samples in almost all cases of comparison, t-test may not be the most suitable. Accordingly, ANOVA needs to be used.

- Language and readability

o There are some typos in the manuscript (e.g., anti-DesRed)

o Some sentences are a bit too long and therefore difficult to read (e.g., The rational is that if Tll suppresses Ase…lack of suppression of Gal4 by Tll)

**Have all data underlying the figures and results presented in the manuscript been provided?**

Reviewer #1: Yes

Reviewer #2: Yes

PLOS authors have the option to publish the peer review history of their article (what does this mean?). If published, this will include your full peer review and any attached files.

Reviewer #1: No

Reviewer #2: No

---

## [Decision Letter · Decision Letter 1]

20 Jan 2022

Dear Dr Zhu,

We are pleased to inform you that your manuscript entitled "The Ets protein Pointed P1 represses Asense expression in type II neuroblasts by activating Tailless" has been editorially accepted for publication in PLOS Genetics. Congratulations!

Yours sincerely,

Hongyan Wang, Ph.D.

Associate Editor

PLOS Genetics

Gregory P. Copenhaver

Editor-in-Chief

PLOS Genetics

Comments from the reviewers (if applicable):

Reviewer's Responses to Questions

**Comments to the Authors:**

Reviewer #1: I am satisfied with the revisions.

Reviewer #2: I am satisfied with the revision.

**Have all data underlying the figures and results presented in the manuscript been provided?**

Reviewer #1: Yes

Reviewer #2: None

PLOS authors have the option to publish the peer review history of their article (what does this mean?). If published, this will include your full peer review and any attached files.

Reviewer #1: No

Reviewer #2: No

**Data Deposition**

http://datadryad.org/submit?journalID=pgenetics&manu=PGENETICS-D-21-01458R1

**Press Queries**

---

## [Editor Report · Acceptance letter]

28 Jan 2022

PGENETICS-D-21-01458R1 

The Ets protein Pointed P1 represses Asense expression in type II neuroblasts by activating Tailless 

Dear Dr Zhu, 

We are pleased to inform you that your manuscript entitled "The Ets protein Pointed P1 represses Asense expression in type II neuroblasts by activating Tailless" has been formally accepted for publication in PLOS Genetics! Your manuscript is now with our production department and you will be notified of the publication date in due course.

With kind regards,

Zsofia Freund

PLOS Genetics

On behalf of:
